# Investigating the link between mineral dust hematite content and intensive optical properties by means of lidar measurements and aerosol modelling

Sofía Gómez Maqueo Anaya[1], Dietrich Althausen[1], Julian Hofer[1], Moritz Haarig[1], Ulla Wandinger[1], Bernd Heinold[1], Ina Tegen[1], Matthias Faust[1], Holger Baars[1], Albert Ansmann[1], Ronny Engelmann[1], Annett Skupin[1], Birgit Heese[1], and Kerstin Schepanski[2]

[1]Leibniz Institute for Tropospheric Research (TROPOS), Leipzig, Germany
[2]Free University of Berlin, Berlin, Germany

**Correspondence:** Sofía Gómez Maqueo Anaya (maqueo@tropos.de)

**Abstract.** This study investigates the relationship between lidar-measured intensive optical properties of Saharan dust and simulated hematite content, using data collected during the Joint Aeolus Tropical Atlantic Campaign (JATAC) in 2021 and 2022. Measurements were taken in Mindelo, São Vicente, Cabo Verde. The study aims to determine how changes in hematite content influence the intensive optical properties of dust particles, particularly in the ultraviolet-visible (UV-Vis) spectrum. Given the well-documented impact of hematite on the absorption properties of dust, especially in the UV-Vis range, our hypothesis is that these effects will be detectable in lidar measurements. Specifically, this study focuses on the lidar ratio, particle depolarization ratio and backscatter- and extinction-related Ångström exponents at 355 nm and 532 nm wavelengths. By analyzing dust plume cases separately regarding their size differences, the strongest positive correlation was identified between the backscatter-related Ångström exponent and hematite fraction ($R^2 = 0.63$). These findings contribute to improving the representation of dust in atmospheric models and refining calculations of its direct radiative effect, which often overlook the variability in mineralogical composition in their dust descriptions.

## 1 Introduction

Mineral dust aerosols are present all around the world. They contribute significantly to the global and regional aerosol loading (Weinzierl et al., 2017), and correspond to a big part of the atmospheric aerosol burden by mass (Kok et al., 2017). Specifically, the Sahara Desert and the Sahel contributes around 50% of the global dust emissions and mass loading (Kok et al., 2021). During transport, dust interacts with the atmosphere in a variety of ways. It modifies the energy balance of Earth through multiple mechanisms, each producing a radiative effect. These impacts can be instantaneous, such as dust scattering and absorption radiation, or may take time to adjust, such as dust altering cloud covers (Boucher et al., 2013). The immediate radiation interactions are well studied; it is known that dust absorbs and scatters solar shortwave and terrestrial longwave radiation depending on its composition (Kok et al., 2017). These interactions have the potential to modify the atmospheric radiation balance from meso- to macro-scale (Kok et al., 2023; Mahowald et al., 2010; Li et al., 2024). However, the magnitude

and sign of this dust radiative effect is still uncertain (Highwood and Ryder, 2014; Kok et al., 2017), one of the major sources of this uncertainty is insufficient knowledge regarding dust absorption properties (Balkanski et al., 2007; Di Biagio et al., 2020; Go et al., 2022). Since dust aerosols are a complex assemblage of various minerals with their distinct physicochemical properties (Formenti et al., 2011), differences in their atmospheric radiative impact will arise as a consequence of distinct mineralogical content.

In order to predict and model the direct radiative effect of mineral dust, vertical profiles of its optical properties are part of the basic necessary input. Among these, the profile of the extinction-to-backscatter ratio, known as the lidar ratio ($S$), is particularly critical (Veselovskii et al., 2020). The lidar ratio, being an intensive optical property (independent of aerosol quantity), is commonly used to characterize mineral dust. However, it has a complex dependency on three major parameters: size, shape and composition (Wandinger et al., 2023). Several studies have aligned differences in dust optical properties to distinct mineralogical content according to source regions (Di Biagio et al., 2017, 2019; Lafon et al., 2004; Sokolik and Toon, 1999; Wagner et al., 2012). Some studies have specifically linked the changes of $S$ to different source regions (Esselborn et al., 2009) and even further to the changes that these differences cause on the dust complex refractive index (CRI) (Go et al., 2022; Schuster et al., 2012; Veselovskii et al., 2020).

In particular, when iron oxides are present in dust particles, the imaginary part of the CRI changes across the radiation spectrum, with an increase in the UV-Vis range. Consequently, they are considered as effective absorbers of the shortwave radiation (Di Biagio et al., 2019; Moosmüller et al., 2012; Sokolik et al., 1993; Sokolik and Toon, 1996; Wagner et al., 2012; Zhang et al., 2024). Furthermore, even small changes in the imaginary part of the CRI (on the order of $10^{-3}$) can strongly affect backscattering efficiency in the UV-Vis spectral region (De Leeuw and Lamberts, 1987; Miffre et al., 2020; Chang et al., 2024). These effects are consistent with the differences observed by Di Biagio et al. (2019), who demonstrated that increasing the imaginary part of the CRI from 0.0016 to 0.0048, associated with a change in hematite fraction of 0.5%, can reduce the single-scattering albedo (SSA) from 0.95 to 0.75 at 370 nm.

Despite this sensitivity, most characterizations of dust based on lidar measured optical properties treat it as homogeneous, giving way to unique wavelength dependent lidar ratio values that indicate "pure Saharan dust". (Haarig et al., 2017, 2022; Groß et al., 2011, 2015; Müller et al., 2013; Schuster et al., 2012; Tesche et al., 2009a, 2011). These studies specifically suggests that the intensive optical properties of dust show no clear regional spectral dependency, despite observed regional spectral differences having been linked to changes in the CRI, specially for the lidar ratio and the backscatter-related Ångström exponent (Schuster et al., 2012; Veselovskii et al., 2020).

The sensitivity study by Veselovskii et al. (2020) demonstrated the impact of changes of the imaginary part of the CRI, due to different iron oxide contents in Saharan dust, on the backscatter-related Ångström exponent for the UV-Vis wavelengths. This relationship is further explored in the laboratory studies by Miffre et al. (2020, 2023), which included particle shape and revealed the non-linearity of the three key physical parameters, size, shape, and composition, affecting the intensive optical properties of dust. Specifically they measured significant impacts of the changes in the complex refractive index for both the backscatter-related Ångström exponents and the particle depolarization ratio for the UV-Vis wavelength range. Complementing these findings, recent modeling efforts have further reinforced the significance of the impact on increasing imaginary parts of

the CRI on dust optical behavior. In the comparative analysis, Chang et al. (2024) evaluated multiple scattering models, including spherical (Mie theory), spheroidal (Dubovik et al., 2006), and irregular hexahedral particle shapes (Saito and Yang, 2021; Saito et al., 2021), and assessed how changes in the imaginary part of the CRI affect the backscatter coefficient at 532 nm, and the backscatter-related Ångström exponent for 355 nm and 532 nm. Although the absolute values differed among the models due to particle geometry, the functional form of the relationship, where increasing imaginary CRI led to decreasing backscatter remained consistent. This convergence across scattering models supports the robustness of using spectral trends derived from simplified assumptions (e.g., spherical particles) as a first-order approximation for interpreting wavelength-dependent backscattering behavior, particularly via intensive optical properties such as the backscatter-related Ångström exponents.

However, several limitations must be considered. Miffre et al. (2020, 2023) conducted their studies in a laboratory setting, which findings may not directly translate to atmospheric conditions, as other atmospheric constituents can have an impact on the measurements. In Veselovskii et al. (2020)'s study, measurements were taken near to an active dust-emitting area, making the presence of larger particles highly likely. For the intensive optical properties where an impact of the complex refractive index has been documented, both, the backscatter-related Ångström exponents and the lidar ratio are particularly sensitive to changes in particle size. Which impact are illustrated by Fig. 1 in Miffre et al. (2020) backscatter-related Ångström exponents and by the Fig. 5 in Wandinger et al. (2023) and Fig. 15 in Zhang et al. (2024) for the lidar ratio. Particle size, shape, and composition influence the measurements, making it challenging to isolate the effects of each factor. Additionally, the lidar measurements conducted by Veselovskii et al. (2020) were taken during a season well known for increased biomass burning activity, leading to frequent encounters with polluted dust in the atmosphere. This pollution alters the measured optical properties, particularly in the UV-Vis spectrum (Heinold et al., 2011; Müller et al., 2009; Tesche et al., 2011), adding another layer of complexity in disentangling compositional effects on both the backscatter-related Ångström exponents and the lidar ratio.

The transport of mineral dust from the Sahara Desert towards the Atlantic Ocean is well-documented (Schepanski et al., 2009, 2017; Tegen et al., 2013; Wagner et al., 2016). During the northern hemispheric (NH) winter months, polluted dust is observed at near-surface layers, complicating the distinction between dust and other aerosols, such as biomass burning aerosols, in lidar measurements. In contrast, during NH summer months, dust is elevated in the atmosphere (Schepanski et al., 2009). Dust transport for the NH summer months is above the marine boundary layer, i.e., on the free troposphere which in turn means that dust transport from the Sahara Desert towards the Atlantic Ocean is more direct. Lidar measurements taken during this time show that the elevated dust layers typically contain lower levels of pollution from other aerosols (Esselborn et al., 2009; Freudenthaler et al., 2009; Haarig et al., 2017, 2019; Groß et al., 2015; Tesche et al., 2009b). Therefore, this study focuses on lidar-measured cases from a remote site in Cabo Verde, to avoid interference from freshly emitted large dust particles, along the mineral dust transport pathway from the Sahara Desert to the Atlantic Ocean. These lidar measurements were taken in Mindelo, São Vicente, Cabo Verde (16°52'39.9"N, 24°59'42.3"W) during NH summer months, from June to September of 2021 and 2022, as part of the Joint Aelous Tropical Atlantic Campaign(s) (JATAC).

This study investigates the potential to identify dust mineralogy within lidar measured optical properties. By analyzing years of continuous, 24/7 lidar measurements of multiple vertical dust profiles above the marine boundary layer, we aim to correlate lidar derived intensive optical properties of dust particles with modeled iron oxide content from an atmospheric model. Given

that previous studies demonstrated the ability of iron oxides on modifying the imaginary part of the CRI, and the impact of those differences on dust intensive optical properties, particularly for the UV-Vis specturm (Di Biagio et al., 2019; Miffre et al., 2020, 2023), this work hypothesizes that this effect will manifest in the lidar measured intensive optical properties at

355 nm and 532 nm wavelengths. The aim is to investigate how the iron oxide content, which varies significantly across the Sahara Desert (Formenti et al., 2011, 2014; Scheuvens et al., 2013), affects dust's intensive optical properties. While regional differences in mineralogy within the Sahara Desert are well-known, few atmospheric models account for mineralogy in their mineral dust descriptions (Chatziparaschos et al., 2023; Gonçalves Ageitos et al., 2023; Li et al., 2021; Menut et al., 2020; Pérez García-Pando et al., 2016; Perlwitz et al., 2015a, b; Solomos et al., 2023).

On a previous study, Gómez Maqueo Anaya et al. (2024) introduced the global, GMINER (Nickovic et al., 2012), mineralogical dataset into the emission code of the aerosol-atmosphere model COSMO-MUSCAT. The configuration presented in that study is what will be used here. GMINER provides spatial distribution of minerals, including iron oxides minerals, based on the methodology of Claquin et al. (1999). This considers an approximate relation of soil mineral fractions to different soil types by taking into account the size distribution, the chemistry and the color of the soil according to the FAO74 classification (FAO-

UNESCO, 1974). However, this approach incurs in a handful of misrepresentations when compared to in-situ measurements (Li et al., 2024; Gonçalves Ageitos et al., 2023). Despite the limitations, Gonçalves Ageitos et al. (2023) found that GMINER dataset fairly reproduces the hematite mass content for the Sahara Desert region (see Fig.11(a) from Gonçalves Ageitos et al., 2023). In GMINER, iron oxide minerals are grouped under the name hematite, and so, we will use hematite and iron oxide interchangeably in this paper. Even though, it is noteworthy that the goethite mineral also contains iron oxide and interacts dif-

ferently with radiation (Di Biagio et al., 2019; Formenti et al., 2014; Go et al., 2022; Wagner et al., 2012), and iron oxides may not always appear as externally mixed aerosols as represented here (Lafon et al., 2004; Kandler et al., 2009). This study combines the modeled hematite content with the lidar retrieved intensive optical properties to investigate potential dependencies between them.

This paper has the following structure: The methodology begins with a general description of the COSMO-MUSCAT model

(Sect. 2.1), followed by a general overview of the lidar system, Polly[XT] (Sect. 2.2). This is succeeded by an explanation of the data selection (Sect. 2.3) and a description of the comparisons between lidar and model dust layers (Sect. 2.4). The section concludes with a brief outline of the POLIPHON method, which converts lidar measurements into dust mass concentrations (Sect. 2.5). In the results section, a case example is shown in Sect. 3.1 where the vertically resolved optical properties are shown in Sect. 3.1.1 followed by a comparison of the POLIPHON and COSMO-MUSCAT results in Sect. 3.1.2. All case studies are

then presented and discussed in Sect. 3.2, the intensive optical properties dependency with hematite is further explored in Sect. 3.2.1 and a separate analysis of the cases due to their size differences is shown in Sect. 3.2.2. Finally, Sect. 4 provides a summary of the paper and findings and discusses implications for future work.

## 2 Methodology

### 2.1 COSMO-MUSCAT

COSMO-MUSCAT is a mesoscale atmospheric model system integrated by two online coupled models. COSMO, developed by the German Weather Service (DWD) is a regional forecast model (Baldauf et al., 2011), while MUSCAT is a chemistry transport model that calculates the atmospheric advective transport of aerosols driven by the forecast model (Heinold et al., 2016; Wolke et al., 2012). The meteorological data are updated every 3 hours and the model runs are reinitialized in overlapping cycles every 48 hours. The use of COSMO-MUSCAT regarding the simulation of mineral dust for the Sahara Desert region

was been thoroughly validated (Gómez Maqueo Anaya et al., 2024; Heinold et al., 2011; Schepanski et al., 2009, 2016, 2017; Tegen et al., 2013). COSMO-MUSCAT is setup for simulating only the transport of Saharan mineral dust including mineralogy within a domain constrained by the following coordinates: 30.75°W, 38.49°N – 39.32°E, 0.38°S. The horizontal grid spacing is 0.25° (28 km) and the vertical resolution contains 40 levels, with a layer thickness of 20 m above sea level for the first layer. The thickness of the subsequent vertical layers varies according to pressure levels, varying from 200 m thickness at 1 km height

to 600 m thickness at 4 km height and a maximum altitude of 21.75 km. The output variables are given for every hour inside of the simulation time. These temporal and spatial resolutions imply a very big difference when compared to a fixed lidar point measurement which is able to retrieve aerosol signals with a significantly higher vertical and temporal resolutions. The difference of these resolutions has to be taken into account when analyzing the results, specially considering that the model gives average dust mass concentration values for the whole São Vicente island.

The mineralogy inclusion is done by incorporating the GMINER mineralogical dataset (Nickovic et al., 2012) in the parametrization of the mineral dust aerosol atmospheric life cycle which includes: (1) dust emission following Tegen et al. (2002), (2) aerosol transport (Wolke and Knoth, 2000), and (3) aerosol removal, which includes both dry and wet deposition (Seinfeld and Pandis, 2016; Zhang et al., 2001; Berge, 1997; Jakobsen et al., 1997). Mineral dust aerosols are transported in MUSCAT in five size segregated classes and are considered as passive traces, meaning that there is not chemically aging or

chemical interaction considered in the simulation.

GMINER follows the Claquin et al. (1999) procedure which extrapolates the mineralogical measurements done for soil classes (following the FAO74 classification (FAO-UNESCO, 1974)) and combines them in order to establish world wide mineral fractions with regards with its soil class in two size classes, namely, clay and silt. The approach is extended in order to consider three new soil types and in particular it extends the hematite fraction to cover both clay and silt sizes. Both extensions

result in differences between -0.0007 to 0.08 in the hematite fraction for the studied domain, where the biggest increment of hematite fraction is found in the Sahel following the north-to-south gradient of hematite for the region (Formenti et al., 2014; Scheuvens et al., 2013). It is evident nonetheless that the uncertainties in the GMINER mineralogical dataset induce errors in calculating the hematite fraction per dust layer. From Fig.1(d) in Li et al. (2024), a comparison between mineralogical datasets and measured iron oxides shows that the Claquin et al. (1999) mineralogical dataset under represents the mass fraction of iron

oxides for Northern Africa. However, the GMINER dataset by Nickovic et al. (2012) is based on Claquin et al. (1999), but specifically expands the distribution of the hematite content in such a way that agrees with measurements in both silt and clay

sizes (Kandler et al., 2009; Wagner et al., 2012). Additionally, the extra soil classifications introduced in GMINER lead to more realistic results through modelling studies (Scanza et al., 2015; Perlwitz et al., 2015a, b). Furthermore, a modeling study comparing mineralogical datasets with in-situ mineralogical aerosol measurements (Gonçalves Ageitos et al., 2023) found that

the GMINER dataset better represents the regional variability of emitted iron oxides from North Africa to the Sahel.

While the GMINER dataset does not distinguish between hematite and goethite content, despite both containing iron oxides, their CRI differs between each other in the UV-Vis spectral range (Formenti et al., 2014; Go et al., 2022; Wagner et al., 2012). Chamber studies by Wagner et al. (2012) and Di Biagio et al. (2019) found that goethite has a lower absorption potential than hematite and is weakly correlated with the imaginary part of the dust CRI. Nonetheless, the modeling study by Li et al. (2024)

found that considering the separation of dust iron oxide content between hematite and goethite does not significantly alter the global shortwave direct radiative effect. However, since goethite is a big part of the iron oxide content in West Africa (Formenti et al., 2014; Go et al., 2022) and due to the locality of our study region, the lack of distinction between hematite and goethite in the GMINER dataset could impact our results, as the absorption capacity may vary depending on which iron oxide mineral is present in the atmosphere.

An additional source of error arises from the modeling approach of mimicking the mineralogical soil size distribution to the aerosol size distribution, disregarding that the emission process changes the dust size distribution (Marticorena and Bergametti, 1995; Kok, 2011; Journet et al., 2014). This change in size distribution from soil to aerosol mineralogy was specifically observed in the chamber study by Wagner et al. (2012), highlighting the need to incorporate these size distribution changes for upcoming modeling efforts. Although some modeling approaches consider these size distribution changes (Li

et al., 2024, 2021; Pérez García-Pando et al., 2016; Perlwitz et al., 2015a, b; Scanza et al., 2015), they all consider the emission parametrization based on Kok (2011)'s brittle fragmentation theory. At present, there is no parametrization that calculates the emission of specific minerals based on Marticorena and Bergametti (1995)'s emission scheme, which is the scheme used in COSMO-MUSCAT.

Along with the GMINER mineralogical dataset the following input files are used for dust's atmospheric life cycle simula-

tion: dust activation frequency map derived from MSG-SEVIRI IR channels (Schepanski et al., 2007), soil vegetation from Copernicus Global Land Service (Fuster et al., 2020), soil moisture from the ERA5 land hourly data (Muñoz Sabater and Copernicus Climate Change Service, 2019), aerodynamic roughness length data set (Prigent et al., 2005), and soil particle size distribution obtained from the SoilGrids database (Poggio et al., 2021). More information regarding the model setup can be found at Gómez Maqueo Anaya et al. (2024).

For this study, the output used from the model are the vertical profiles of total and size segregated dust mass concentrations and hematite mass concentrations, from which the hematite fraction is obtained by dividing the hematite mass concentration by the total dust mass concentration. The model results are all from above the grid cell corresponding to São Vicente, Cabo Verde for the simulation periods of 8 August - 30 September 2021 and 2 June - 31 July 2022.

## 2.2 Polly[XT]

The lidar data for this study were obtained from an automated multiwavelength Raman polarization and water-vapor lidar, called Polly[XT] (POrtabLe Lidar sYstem, the XT superscript refers to the updated version) (Althausen et al., 2009; Engelmann et al., 2016). It is part of the Polly[NET], which is a network of Polly systems around the world (Baars et al., 2016). This specific Polly[XT] provides continuous measurements since June 2021 at the OSCM (Ocean Science Center Mindelo) located at Mindelo, São Vicente, Cabo Verde (16°52'39.9"N, 24°59'42.3"W). The produced data from this lidar system has been used for aerosol

characterization (Gebauer et al., 2024) and validation purposes (Baars et al., 2023; Gómez Maqueo Anaya et al., 2024). The Polly[XT] systems emit linearly polarized light pulses at the three wavelengths of 355, 532, and 1064 nm (covering the UV-IR part of the spectrum); the receiver of the system has fifteen channels, three are used for measuring the backscatter light at the emitted wavelengths, three channels detect cross-polarized light at 355, 532, and 1064 nm wavelengths. The additional four receiver channels at 387, 407, 607, and 1058 nm wavelengths are used to detect Raman scattering at nighttime. Signals are

measured with a vertical resolution of 7.5 m (1 bin) and a temporal resolution of 30 s. The lidar observations presented here were manually analyzed, a vertical smoothing is necessary to reduce noise in the measurements and facilitate their interpretation. However, the vertical smoothing has been always applied in such a manner that the retrieved values are representing particle values inside the investigated aerosol layer and are not depending on values from outside the aerosol layer. A thorough system description can be found in Gebauer et al. (2024).

The vertical resolved aerosol optical measurements resulting from the lidar signals can be categorized into two main categories: extensive properties and intensive optical properties. Extensive optical properties, which depend on aerosol concentration, include the particle extinction coefficients (absorption plus scattering) and the particle backscattering coefficients at 355, 532, and 1064 nm wavelengths ($\alpha_{355}$, $\alpha_{532}$, $\alpha_{1064}$ and $\beta_{355}$, $\beta_{532}$, $\beta_{1064}$). Intensive optical properties, which are independent of aerosol concentration, include the particle linear depolarization ratios at 355, 532 nm and 1064 nm wavelengths ($\delta_{355}$,

$\delta_{532}$, $\delta_{1064}$), the extinction-to-backscattering ratio, or lidar ratio, for the UV-Vis wavelengths ($S_{355}$, $S_{532}$) and a possibility for obtaining the IR lidar ratio ($S_{1064}$). The following Ångström exponents are available, backscatter-related, extinction-related and lidar ratio-related for the 355 to 532 nm and for the 532 to 1064 nm wavelength ranges (ÅE$(\beta)_{355/532}$, ÅE$(\beta)_{532/1064}$, ÅE$(\alpha)_{355/532}$, ÅE$(\alpha)_{532/1064}$, and ÅE$(S)_{355/532}$, ÅE$(S)_{532/1064}$). Uncertainties associated with most of these measurements can be found in Table 1 in Hofer et al. (2017), and further discussed in Freudenthaler et al. (2009); Baars et al. (2012, 2016)

and Engelmann et al. (2016). Intensive optical properties commonly used for dust characterization as the single scattering albedo or the mass absorption coefficient (Di Biagio et al., 2019; Zhang et al., 2024) cannot be directly obtained from lidar measurements.

The conversion of received signals into optical measurements can generally be done by following one of two methods: the Klett method (Klett, 1985) or a combined Raman elastic-backscatter approach. The Klett method requires an initial estimate of

the lidar ratio for the retrievals. In contrast, the Raman method does not require this initial guess, as it allows the independent determinations of the extinction and the backscatter coefficients (Ansmann et al., 1992). Since our focus is on independent lidar ratio measurements, we use only the Raman methodology. However, Raman lidar applications are limited to nighttime

measurements because of the necessary inelastic-backscattering signal that can only be detected when the strong daylight background is not there (Ansmann et al., 1992).

Intensive optical properties are employed for aerosol characterization. Lidar ratios and particle linear depolarization ratios are dependent on particle size, shape, and composition (Hofer et al., 2020; Huang et al., 2023; Miffre et al., 2023; Saito and Yang, 2021; Schuster et al., 2012; Wandinger et al., 2023; Veselovskii et al., 2020). While the Ångström exponents are initially believed to be primarily related to the particle size (Ångström, 1929), laboratory and observational studies suggest a substantial dependence of the backscatter-related Ångström exponent on composition, especially when changes in composition lead to changes in the imaginary part of the CRI at UV-Vis wavelengths (Miffre et al., 2020; Veselovskii et al., 2020). Under varying conditions, changes in these dependencies affect the optical properties differently, portraying nonlinear relationships between particle physical characteristics and their optical properties.

## 2.3 Data selection

The selection of aerosol layers intended to study the relation of lidar-measured properties and modeled hematite content focuses on identifying cases where mineral dust was the dominant aerosol. This selection process involves three main, consecutive steps: (1) utilizing AERONET (Aerosol Robotic Network; Holben et al., 1998) measurements to identify potential dust layers, (2) analyzing available Polly$^{XT}$ measurements to confirm these dust layers, and (3) reviewing COSMO-MUSCAT simulation output for the identified cases. An overview of this selection process is illustrated by Fig. 1.

The selection process is constrained by two specific time periods from NH summer measurements campaigns centered around Cabo Verde. The JATAC campaigns, which took place from June to September in both 2021 and 2022, used ground-based, including the Polly$^{XT}$, aircraft, and balloon measurements to validate data provided by ESA's Aeolous satellite. We decided to focus on these campaigns periods for two main reasons: First, in the spirit of the campaigns' objectives, the data was constantly quality controlled and cross-checked between measurement devices. Second, the summer months are ideal due to the seasonality of Saharan dust transport towards the Atlantic Ocean. During these months, dust travels the highest in the atmosphere (Schepanski et al., 2009) with less interference from other aerosols. Given the data availability of the Polly$^{XT}$ lidar, we narrowed our focus to the following periods for identifying dust-laden aerosol layers: August - September 2021 and June - July 2022.

Within the selected time frame, the next step involved identifying days with dust-dominated aerosol layers. As a first approximation, the total column optical measurements recorded by the AERONET sun-photometer in Mindelo, Cabo Verde, were used. The AERONET level 2.0 (quality-assured and cloud-screened) dataset was then filtered based on the following "pure dust" criteria: Aerosol Optical Thickness for the 550 nm wavelength ($AOT_{550}$) > 0.1 and the exctinction-related Ångström exponent for the 440-870 nm wavelengths ($\text{ÅE}(\alpha)_{440/870}$) < 0.3 (Ansmann et al., 2019). As a result of the filtering process, only the dates meeting the "pure dust" criteria remained for the subsequent steps of the selection process.

The second step of the selection process involved analyzing Polly$^{XT}$ data. The initial task was to verify whether the dates that passed the AERONET related filters included nighttime lidar measurements. The next step was the manual cloud-screening procedure, as the presence of clouds can influence all particle optical properties. If retrievals can be conducted without cloud

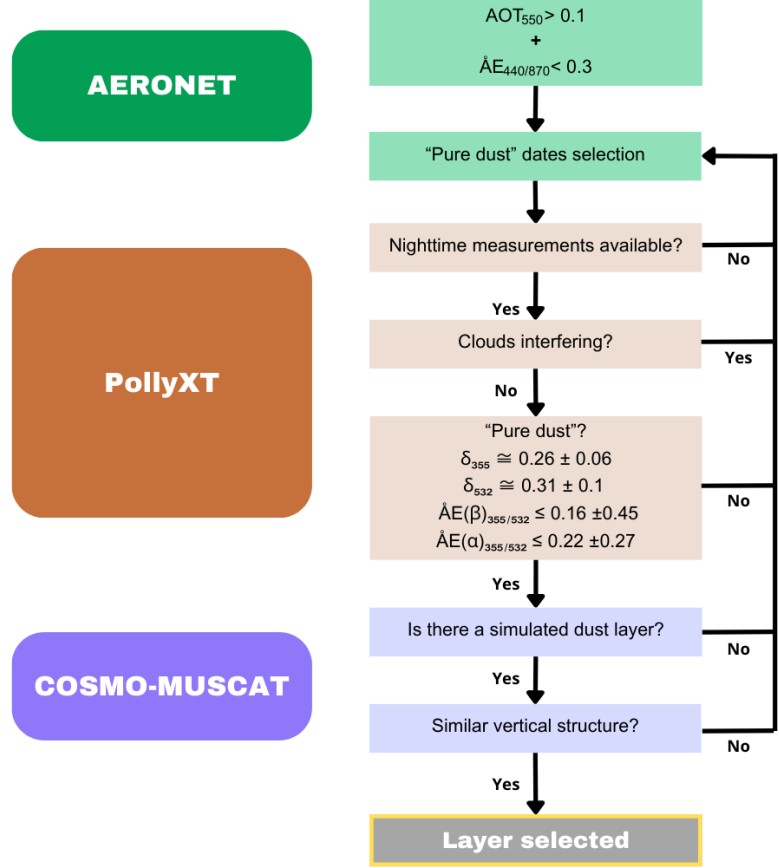

**Figure 1.** Flow chart of the case selection process. Dust layers are selected through three main steps, the first step is depicted with green colors and relates to date filtering through AERONET measurements and the so-called "pure dust" optical measurements values (Ansmann et al., 2019). The second step is through Polly[XT] measurements, shown in the figure in orange colors. The first two filters surround the data availability and if clouds are interfering with the measurements while the last filter regards the lidar retrieved optical measurements (Tesche et al., 2011). The third step is pictured in violet colors and it is related to the simulation results from COSMO-MUSCAT, the questions to answer in this section are related to if the dust layer is simulated in a similar way to the lidar vertical structure. Acronyms: Aerosol Optical Thickness at 550 nm wavelength ($AOT_{550}$), extinction-related Ångström exponent for the 440-870 nm wavelengths ($\text{ÅE}(\alpha)_{440/870}$), particle linear depolarization ratio at the 355 nm wavelength ($\delta_{355}$), particle linear depolarization ratio at the 532 nm wavelength ($\delta_{532}$), backscatter-related Ångström exponents for the 355-532 nm wavelengths ($\text{ÅE}(\beta)_{355/532}$), and extinction-related Ångström exponent for the 355-532 nm wavelengths ($\text{ÅE}(\alpha)_{355/532}$).

interference, or if specific time periods minimize the impact of clouds, the selection process proceeded with those suitable times for dust retrievals. When clouds are located below the aerosol layer, they interfere with lidar retrievals targeting the aerosol plume above. This is because multiple scattering within clouds prevents accurate measurement of the aerosol plume.

Conversely, if clouds are above the aerosol layer, they do not necessarily contaminate aerosol signals, provided there is sufficient separation between the clouds and the aerosol layer. This separation needs to be sufficiently distinct to allow for Raman elastic-backscatter retrievals using a reference height in that intermediate space.

From this point forward, the selection of "pure dust" days and the nighttime hours from which the vertical optical properties of dust are retrieved from the Polly$^{XT}$ measurements has been made. Some "pure dust" days were found to have two distinct

dust aerosol layers. Each layer is considered as an individual case study because such layering could be due to atmospheric inversions or to dust originating from different source regions with their own wind pattern, potentially affecting the modeled hematite content between layers. The subsequent step involves selecting dust layers that meet the "pure dust" criteria based on lidar optical measurements, excluding considerations for lidar ratios. The criteria, based on Tesche et al. (2011), are as follows: $\delta_{355} \simeq 0.26 \pm 0.06$, $\delta_{532} \simeq 0.31 \pm 0.1$, $\text{ÅE}((\beta)_{355/532}) \leqslant 0.16 \pm 0.45$, $\text{ÅE}((\alpha)_{355/532}) \leqslant 0.22 \pm 0.27$ as illustrated in the last

subsection of the Polly$^{XT}$ step in Fig. 1.

The third step of the selection process takes into account the COSMO-MUSCAT modelling results. The first criterion is whether the model simulates the dust layer(s) for the same date and time where the dust layer(s) were observed by the Polly$^{XT}$. If this criterion is met, the next step is to assess if the model reproduces a similar vertical structure. A dust layer is selected for this study if the model both simulates the dust layer(s) at the corresponding dates and times and simulates a comparable

vertical structure. Between 19-08-2021 and 08-07-2022, 22 dust layer cases passed the data selection filters.

## 2.4 Layer comparison

The comparison between the vertically resolved optical property, the lidar ratio, at both 355 nm and 532 nm wavelengths and the modeled hematite fraction is illustrated in Fig. 2 for a hypothetical case based on lidar and model data. This comparison is done for each dust layer, where a mean value is calculated based on their own vertical structure. Specifically, the optical

properties means are calculated by taking into account the thicknesses of each dust layer as retrieved by Polly$^{XT}$, while the hematite fraction means are calculated based on the thickness of each COSMO-MUSCAT simulated dust layer. For instance, for the two dust layers sketched in the Fig. 2, one mean lidar ratio value at 355 nm and another at 532 nm are calculated for the dust layer in between 1.4 and 3 km. These values are then compared to the mean hematite fraction (hematite mass concentration divided by the total dust mass concentration) for the simulated dust layer in between 0.9 and 2.8 km. For the lofted layer, the

lidar ratio means are computed for the range between 3.7 and 5.2 km and are compared with the modeled mean hematite fraction in the range of 3 to 5.3 km.

## 2.5 POLIPHON

The two-step POLIPHON method (Mamouri and Ansmann, 2014, 2017) translates lidar-retrieved optical properties into mass concentrations of coarse dust (particle diameter (D) > 1 μm), fine dust (D < 1 μm), and non-dust concentrations. This separation

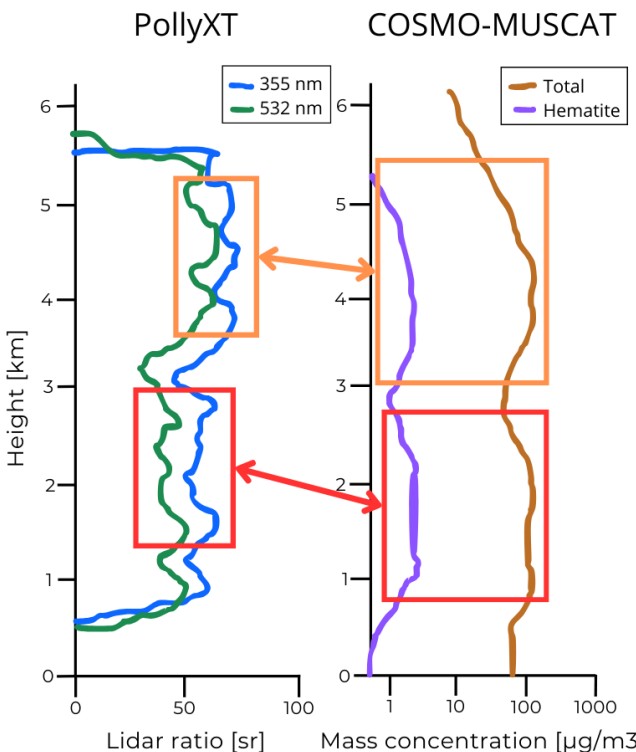

**Figure 2.** Sketch illustrating how the dust layers are compared between the Polly[XT] results and the COSMO-MUSCAT simulated dust layers. On the left hand side the lidar ratio vertical structure is sketched for the 355 nm (blue) and 532 nm (green) wavelengths. On the right hand side, the vertical structure of the modeled total dust mass concentration (brown) and hematite mass concentration (purple) are sketched. The boxes represent the domain of one dust layer and the arrows represent relation between the measurements. The sketch represents dust related vertically distributed properties above Mindelo, São Vicente, Cabo Verde.

is achieved by utilizing the lidar retrieved $\delta$ and $\beta$ values, AERONET-derived conversion factors, and assumed values for $S$ and density. The method involves the separation of the measured $\beta$ values using the retrieved $\delta$ values to estimate $\alpha$ values based on assumed $S$ values, and afterwards, converting the calculated particle extinction coefficients into mass concentrations by applying the AERONET-derived conversion factors and an representative densities ($\rho$). The specific values used for the dust layers analyzed in this study are provided in Table 1. The two-step POLIPHON processes are described in detail below.

**Separation of particle backscatter coefficient:** this procedure consists of two processes. Initially, a broad separation between coarse dust and the mixture of fine dust and non-dust aerosols is performed based on $\delta$ values (Fig. 3b). Defined threshold values of $\delta$ are compared to the lidar retrieved $\delta$ values to achieve this separation. For this first step, the $\delta$ values for coarse dust, as listed in Table 1, is applied, together with the fine dust and non-dust mixture $\delta$ value. In this study, a $\delta$ value of 0.12 is used for the mixture of fine dust and non-dust, as the analyzed dust layers are located above the marine boundary layer, where non-dust aerosols are primarily of a continental nature (Mamouri and Ansmann, 2017).

**Table 1.** Used values for implementing the two-step POLIPHON method. Values are obtained from Ansmann et al. (2019, 2012); Mamouri and Ansmann (2017). $\delta$ is the particle depolarization ratio, $S$ is the lidar ratio and $C_v$ is the volume conversion factor, and $\rho$ represents density. Units in square brackets.

| Parameter | Non-dust | Dust | Coarse dust | Fine dust |
|---|---|---|---|---|
| $\delta$ | 0.05 | 0.31 | 0.39 | 0.16 |
| $S$ | 50 | 50 | 50 | 50 |
| $C_v$ [$10^{-12}$Mm] | 0.22±0.06 | 0.64±0.07 | 0.79±0.08 | 0.22±0.06 |
| $\rho$ [g/cm$^3$] | 1.5 | 2.6 | 2.6 | 2.6 |

The second process separates fine dust from non-dust aerosols using $\delta$ threshold values fo these categories, as shown in Table 1. Ultimately, the values of $\beta$ for coarse dust, fine dust, and non-dust are determined. Then $\delta$ threshold values used for the separation of non-dust and dust categories are derived from extensive lidar measurements (Mamouri and Ansmann, 2014), while the coarse and fine dust $\delta$ values are obtained from laboratory studies (Sakai et al., 2010 and Järvinen et al., 2016, respectively). Additionally, the $\delta$ values assigned to the fine dust and non-dust mixture depend on the nature of the non-dust aerosol and is determined iteratively by comparing the results it yields with lidar measurements, as detailed in Mamouri and Ansmann (2014, 2017).

**Calculation of particle extinction coefficient:** once the backscatter coefficients are separated, $\alpha$ is calculated by applying appropriate lidar ratio values for each fraction. The same $S$ values are assumed for dust-related fractions while the non-dust value is based on the likelyhood of the aerosols origin. The estimates are based on regional data and are derived from an extensive collection of previous lidar observations (Mamouri and Ansmann, 2014, 2017). The extinction coefficient is obtained by multiplying the separated $\beta$ by their corresponding $S$ values.

**Conversion to mass concentrations:** the extinction coefficients are then converted into volume concentrations using specific conversion factors, followed by the transformation into mass concentrations through the application of appropriate density values. These conversion factors are derived from AERONET aerosol climatologies, while the representative densities are based on previous aerosol studies and the assumption of homogeneity within dust and non-dust categories (Ansmann et al., 2012, 2019).

The POLIPHON method incorporates several assumptions, the most notable being the uniformity of optical properties across all mineral dust particles, disregarding heterogeneity due to different source regions. However, a significant advantage of this approach is that it does not require a dust particle shape model for data analysis, relying solely on the measured optical properties. Nevertheless, since the method involves a conversion factor based on AERONET climatologies, it is important to note that in order for AERONET to derive aerosol mass fractions, a spheroid shape model is used (Dubovik et al., 2006). The two-step POLIPHON method has been validated, demonstrating good agreement between airborne measurements and lidar retrieved, two-step POLIPHON derived, fine and coarse modes of dust mass concentrations on Barbados during the SALTRACE campaign (Haarig et al., 2019).

The two-step POLIPHON derived data are employed in this study as an analytical tool with two key mass concentration ratios serving specific purposes. First, the ratio of non-dust to dust mass concentration is used to color-code each dust layer case and for quantifying relationships between different case studies. Second, the fine-to-coarse dust mass concentration ratio is used to categorize the case studies into three distinct groups: (1) dust layers with a higher portion of fine dust mass concentration (i.e., fine/coarse > 11%), (2) dust layers with an intermediate proportion of fine dust mass concentrations (i.e., 9% < fine/coarse < 11%), and (3) dust layers with a lower portion of fine dust mass concentration (i.e., fine/coarse < 9%). It is important to emphasize that this classification is specific to the analyzed cases and does not represent a universal standard for defining "high" fine-to-coarse dust mass concentration. Moreover, considering the uncertainties associated with mass concentrations calculations, 20 - 30% for total mass concentration, 40 - 60% for the fine-mode mass concentration, and 25 - 35% for coarse-mode mass concentration (Ansmann et al., 2019), the proposed classification falls outside the uncertainty margin. Therefore, these divisions should be regarded as exploratory and artificial, intended solely for the purposes of this study. The primary objective is to investigate a potential method for disentangling physical parameters influencing intensive optical properties, particularly in light of the evident impact of dust particle size on the lidar ratio, as shown in Fig. 5 in Wandinger et al. (2023) and Fig. 15 in Zhang et al. (2024).

Additionally, to support the validation of model results, the fine-to-coarse dust mass concentration ratio derived from the POLIPHON is compared with the corresponding ratio from COSMO-MUSCAT for each dust layer. This comparison enhances the reliability of the analysis by providing a more robust assessment of model and lidar-derived data agreement. The described products and their application are illustrated in Fig. 3.

## 3 Results and their discussion

This section is structured as follows: First, lidar measurements from 24 August 2021 are shown and described. On this day, two distinct dust layers are identified and analyzed as separate case studies. The dust layers are shown after applying the 2-step POLIPHON method, and the results are compared with the corresponding COSMO-MUSCAT results for the same day and time. Next, the bulk of case studies is presented, beginning with a comparison of the fine-to-coarse ratios derived from COSMO-MUSCAT and POLIPHON. This is followed by an analysis of the relationship between hematite fraction and lidar ratio, as well as particle linear depolarization ratio at both at 355 nm and at 532 nm, and backscatter and extinction-related Ångström exponents. Finally, the cases analyzed separately according to size, the hematite fractions are compared with the aforementioned lidar-measured optical properties.

### 3.1 Single case study - 24 August 2021

#### 3.1.1 Observed dust optical properties

Figure 4 shows the vertical distribution of optical properties retrieved from Polly$^{XT}$ on the 24 of August 2021 between 02:45 UTC and 05:27 UTC. The vertical profiles show two aerosol layers. In order to accurately determine their heights, a

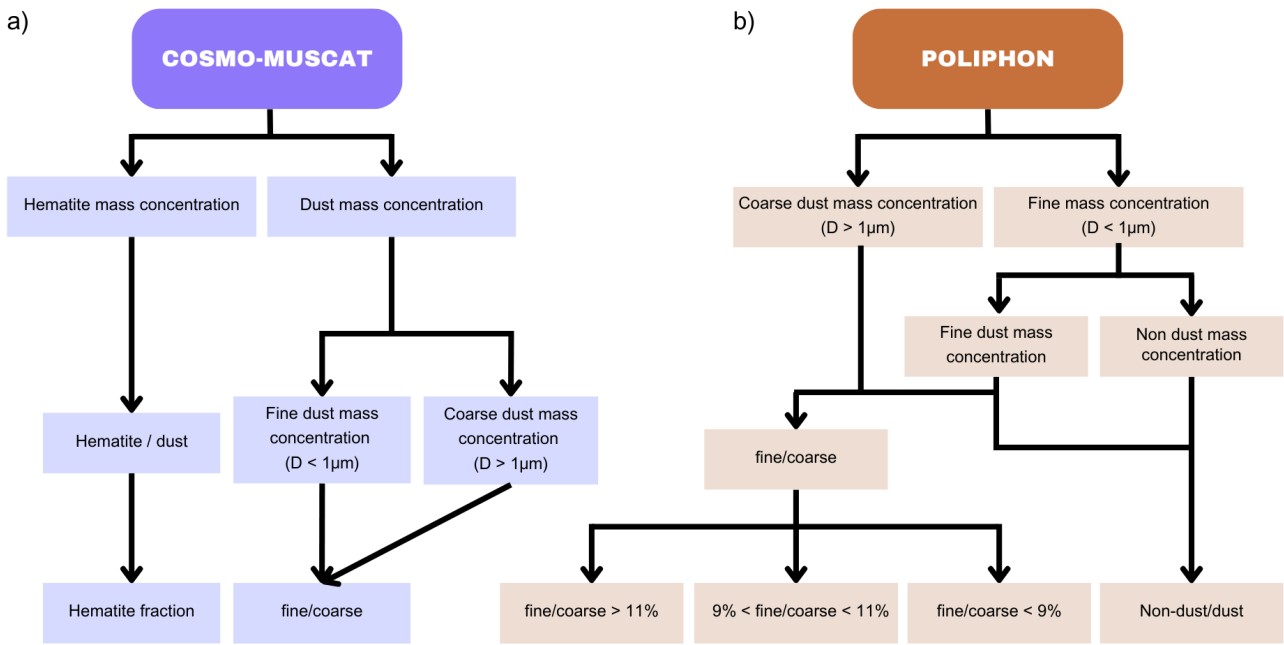

**Figure 3.** Process flow illustrating two main methods and their corresponding data outputs used in the study. (a) Violet colors represent the output from COSMO-MUSCAT. The fine-to-coarse dust mass concentration ratio, and hematite fraction are the model output products used for the results. The particle diameter threshold for distinguishing between fine and coarse dust mass concentration is set at 1 μm to align with the POLIPHON classification. (b) The POLIPHON output flow is shown in orange colors. The products from the POLIPHON used for the results are the fine-to-coarse dust, and the non-dust to dust mass concentration ratios.

lower vertical smoothing is used to enhance the distinction between aerosol layers and air layers with less particle contents. The heights of the layers are then set using a vertical smoothing of 22.5 m. By cross-checking the measured optical properties against the values signaling for dust layers (Tesche et al., 2011) in Cabo Verde (step 2 of Fig. 1's flow chart), it is confirmed that the measurements for this day correspond to two distinct dust layers, each to be considered as a separate case study (see Table 2). The vertical smoothing used to calculate their optical properties is different for the layers to balance the number of independent measurements and the length of their standard variation. For the lower layer near the surface, a vertical smoothing of 232.5 m is applied, while for the upper layer, a vertical smoothing of 577.5 m is used, which profiles are illustrated in Fig. 4. It is noteworthy that the particle backscatter coefficient shows a wavelength dependence, with larger $\beta$ values for 532 nm than at 355 nm for the upper dust layer but not for the lower layer. The mean values of the particle extinction coefficient fall within their measurement error margins when comparing 355 nm to 532 nm wavelengths across both dust layers, implying wavelength independence. Such behavior is not uncommon for Saharan dust (Veselovskii et al., 2016, 2020). Since our study focuses on

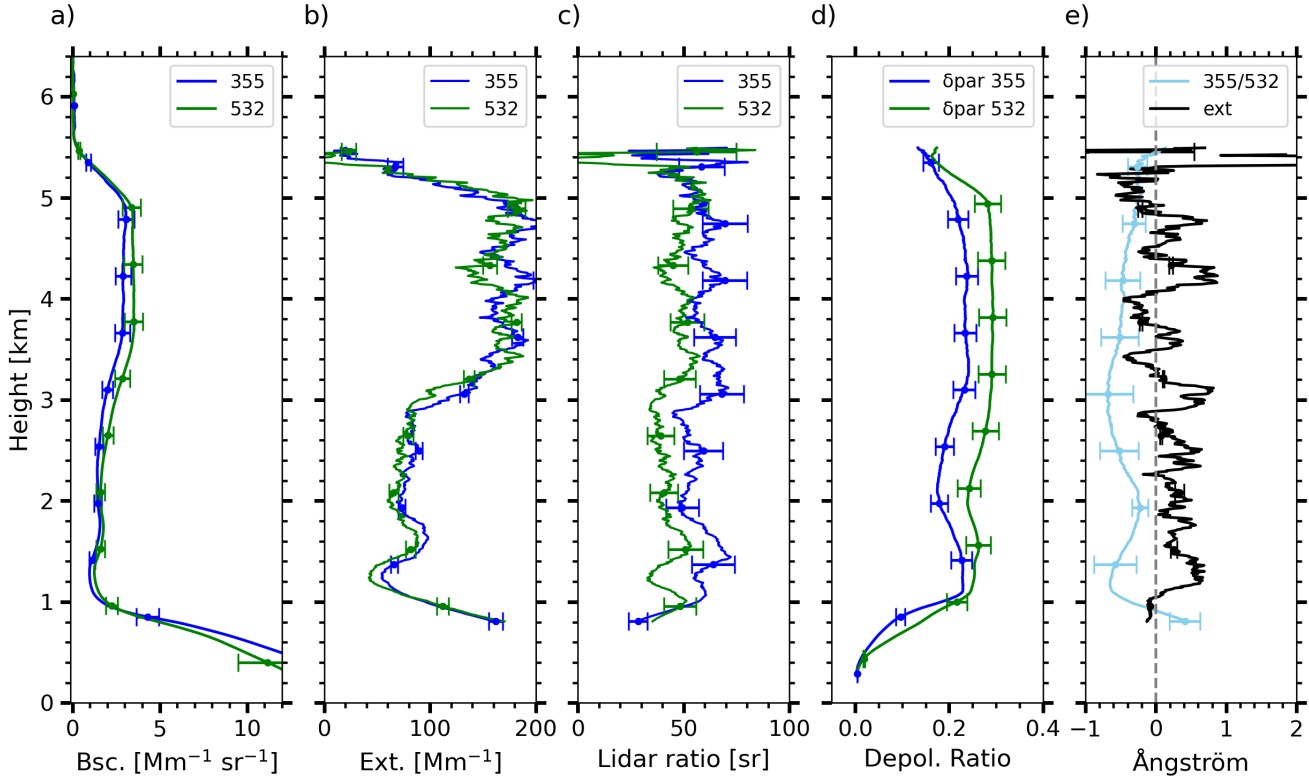

**Figure 4.** Average lidar profiles measured on 24 August 2021, 02:45-05:27 UTC. The vertical profiles are vertically smoothed before calculation with a resolution of 577.5 m. For plots (a-d), blue represents measurements at 355 nm and green represents measurements at 532 nm wavelength. The optical properties shown are: particle backscatter coefficient (Bsc. ($\beta$)) (a), particle extinction coefficient (Ext. ($\alpha$)) (b), lidar ratio ($S$) (c), and particle linear depolarization ratio (Depol. ratio ($\delta$)) (d). For the Ångström exponents (ÅE) plot (e), both ÅE shown are calculated for the 355-532 nm wavelengths, the backscatter-related ÅE is shown in light blue and the extinction-related ÅE is shown in black.

the optical properties of dust independent from the amount of it, with particular interest in backscattering characteristics, the emphasis is in the intensive optical properties. Table 2 provides the mean values of these intensive optical properties for each
dust layer together with Tesche et al. (2011)'s reference "pure dust" values.

The mean values of the intensive optical properties reveal that the lidar ratios at 355 nm fall within the upper half of the "pure dust" tolerance range, whereas the lidar ratios at 532 nm are in the lower half of this range. For the particle linear depolarization ratios, the values at 355 nm are in the lower half of the "pure dust" tolerance range, while the values at 532 nm are within the inner half of the "pure dust" tolerance range. The backscatter-related Ångström exponent values are either on the lower end
(lower layer) or below the lower limit of the specified range (upper layer), whereas the extinction-related Ångström exponent values are either above the upper limit of the "pure dust" range (upper layer) or on the lower limit of the specified range (lower

**Table 2.** Mean values of lidar ratios ($S$), particle linear depolarization ratios ($\delta$), backscatter-related Ångström exponent ($\text{ÅE}(\beta)$) and extinction-related Ångström exponent ($\text{ÅE}(\alpha)$) for the 355 nm and 532 nm wavelengths for two dust layers measured on 24 August 2021, 02:45-05:27 UTC. Reference "pure dust" values from Tesche et al. (2011) on the last row.

| Height | $S_{355}$ [sr] | $\delta_{355}$ | $S_{532}$ [sr] | $\delta_{532}$ | $\text{ÅE}(\beta)_{355/532}$ | $\text{ÅE}(\alpha)_{355/532}$ |
|---|---|---|---|---|---|---|
| 1.4 - 2.1 km | 59±6 | 0.201±0.012 | 44±10 | 0.252±0.007 | -0.26±0.05 | 0.54±0.57 |
| 2.7 - 5.2 km | 61±5 | 0.236±0.003 | 48±4 | 0.292±0.001 | -0.49±0.07 | 0.08±0.34 |
| "pure dust" following Tesche et al. (2011) | 53±10 | 0.26±0.06 | 54±10 | 0.31±0.1 | 0.16±0.45 | 0.22±0.27 |

layer). Values close to zero for both $\text{ÅE}(\beta)$ and $\text{ÅE}(\alpha)$ are related to the presence of large particles. Negative values are not unusual for Saharan dust (Haarig et al., 2022; Veselovskii et al., 2016, 2020), with measurements showing $\text{ÅE}(\beta)$ ranging from -0.55 to 0.5 and $\text{ÅE}(\alpha)$ ranging from -0.2 to 0.2. Negative $\text{ÅE}(\beta)$ values can result from a spectral dependence on changes in the imaginary part of dust's CRI at UV-Vis wavelengths. A sensitivity study performed in Veselovskii et al. (2020) shows that the $\text{ÅE}(\beta)$ is more affected than $\text{ÅE}(\alpha)$ by changes of the imaginary CRI part at the UV-Vis range.

In the present case, the increase of $S$ values, along with the changes in $\text{ÅE}(\beta)$ between lower and upper layers, suggest a compositional differences, likely the presence of a mineral such as hematite in the lower layer, which enhances absorption by increasing the imaginaray part of the CRI, driving the backscattering coefficient to decrease in a spectral dependent manner (Chang et al., 2024; De Leeuw and Lamberts, 1987; Wandinger et al., 2023). The fact that $\text{ÅE}(\beta)$ in the lower layer is closer to zero supports the hypothesis of higher hematite content in this layer. While the $S$ differences fall within the uncertainty range, the contrast in $\text{ÅE}(\beta)$ is more pronounced and may serve as a more sensitive indicator. To better understand the influence of hematite on the spectral backscattering behavior, and given the limitations of single-layer interpretation, we extend our analysis to a broader set of dust cases. This approach enables a more robust assessment of the role of mineral composition, particularly iron oxides, in shaping lidar-derived intensive optical properties.

### 3.1.2 POLIPHON derived vs COSMO-MUSCAT dust concentrations

The results from applying the two-step POLIPHON method to measurements from 24 August 2021, alongside COSMO-MUSCAT simulation results for that same day at a similar time range (3:00-6:00 UTC) are presented in Fig. 5. The COSMO-MUSCAT simulation results depicts a two dust layer structure; however, the altitudes of these layers do not align precisely with those observed. For comparison purposes, the COSMO-MUSCAT dust layers are defined as follows: lower layer extends from 0.8 to 2.7 km, while the upper layer ranges from 3 to 4.8 km.

According to the POLIPHON results, the average dust mass concentration in the lower layer is 118 µg/m$^3$, while the upper layer reaches 270 µg/m$^3$. COSMO-MUSCAT simulation estimate an average dust mass concentration of 179 µg/m$^3$ for the lower layer, and 97 µg/m$^3$ for the upper layer. This indicates that the model overestimates the dust concentration in the lower layer by a factor 1.5 and underestimates it in the upper layer by a factor of 0.35. Despite these discrepancies, the model fits well

to the total measured dust mass load. This can be seen in the good agreement between AERONET and COSMO-MUSCAT total column AOT for the Mindelo station on 23 August 2021 at 19:00 UTC, where the model calculates a total column AOT of 0.84, while AERONET measurements show an AOT of 0.76 (Fig A1).

Regarding the fine-to-coarse dust ratio, POLIPHON derived values indicate an average of 0.15 for the lower layer and 0.08 for the upper layer. COSMO-MUSCAT simulates a ratio of 0.1 for the lower layer and 0.12 for the upper layer. The model simulates coarser particles for the layer closer to the ground, which may explain the higher modeled dust mass concentration for that layer. Conversely, the model estimates a greater fraction of fine dust in the upper layer compared to observed, which may contribute to the underestimation of mass concentration at those altitudes. Despite these discrepancies, the model effectively represents the two-layer dust structure, and the two datasets are sufficiently well related, as the mass concentrations are of the same order of magnitude. This level of agreement allows for a meaningful combined analysis, particularly since the focus of this study is on mass-independent properties. It is noteworthy that the discrepancies may also stem from the model averaging the dust plumes across the entire island, making it inherently unable to capture the exact same vertical structure that is observed by lidar measurements.

Furthermore, the average hematite fraction per dust layer is 0.009 for the lower layer and 0.011 for the upper layer, indicating only a slight difference between the two. This small difference does not support the expectation that the lower layer contains more hematite, as previously considered based on ÅE($\beta$). Additionally, lidar ratio values are nearly identical between layers, making it difficult to attribute any observed optical differences solely to hematite content. However, the more distinct increase in ÅE($\beta$) remains notable and may reflect other influencing factors, such as differences in particle size distribution.

## 3.2 Multiple case studies

Figure 6 shows the fine-to-coarse dust mass concentration ratios obtained from POLIPHON compared to the COSMO-MUSCAT simulation results for 22 case studies. The results show that the ratios are in the same order of magnitude and they all fall within the range of the 2-to-1 and 1-to-2 lines. There is a notable tendency for COSMO-MUSCAT to lean towards the 1-to-2 comparison, indicating a slight overestimation of the fine portion of the dust mass concentration.

Studies by Adebiyi and Kok (2020) and Kok et al. (2017) compared ensembles of atmospheric models with dust aerosol in-situ measurements and found that atmospheric models tend to over predict the fine dust portion while underestimating the coarse dust portion. This over and under prediction has impacts on the calculation of the global direct radiative effect. Fine dust significantly contributes to the extinction at 550 nm, with only a few percent of that extinction due to absorption, whereas coarse dust absorbs a larger fraction of the extinguished radiation at 550 nm (Adebiyi et al., 2023).

To further validate the model results, a comparison of the aerosol mass loading was conducted by analyzing total column aerosol optical thickness (AOT) at 550 nm against AERONET data. This comparison considered five AERONET stations strategically distributed across the Sahara Desert region and along the dust transport pathway over the Atlantic, with corresponding grid cells values in the model results. Despite certain limitations, such as extended periods of missing data for some stations, and specific events being misrepresented by the model, COSMO-MUSCAT effectively reproduces the temporal evolution of

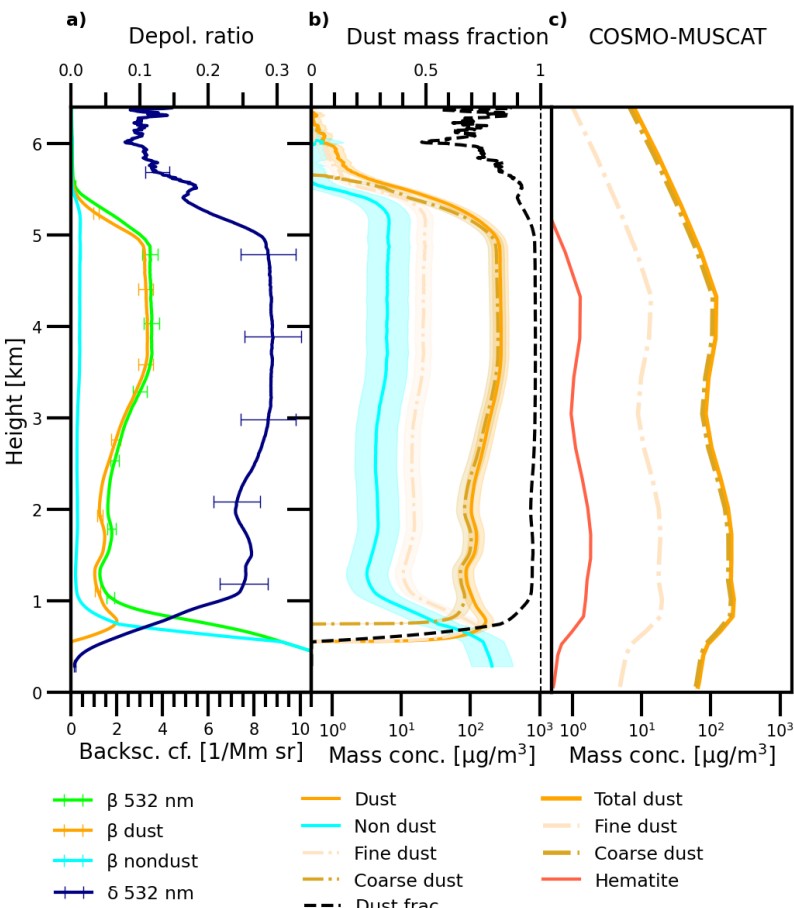

**Figure 5.** Vertical profiles for the POLIPHON derived products from lidar measured optical properties on 24 August 2021 02:45-05:27 UTC, and COSMO-MUSCAT simulation results above Mindelo. The 532 nm particle backscatter coefficient (a, light green) and the particle linear depolarization ratio (a, dark blue) are the input to obtain the separated dust and non-dust profiles in (a, b). The POLIPHON products are the derived 532 nm dust backscatter coefficient (a, orange) and the non-dust backscatter coefficient (a, light blue), dust mass concentration (b, orange), coarse dust mass concentration (b, gold), fine dust mass concentration (b, bisque), the non-dust mass concentration (b, light blue), and the dust mass fraction (b, black, ratio of the dust to total particle mass concentration, dashed black vertical line indicates a dust mass fraction of 1). Vertical profiles of simulated mineral dust mass concentrations from the COSMO-MUSCAT model (c). The vertical profile corresponds to values calculated for the grid cell where Mindelo, Cape Verde is found in the model for 24 August 2021 3:00-6:00 UTC. Total dust mass concentration (c, orange), coarse dust mass concentration (c, gold), fine dust mass concentration (c, bisque), and hematite mass concentration (c, tomato) are shown.

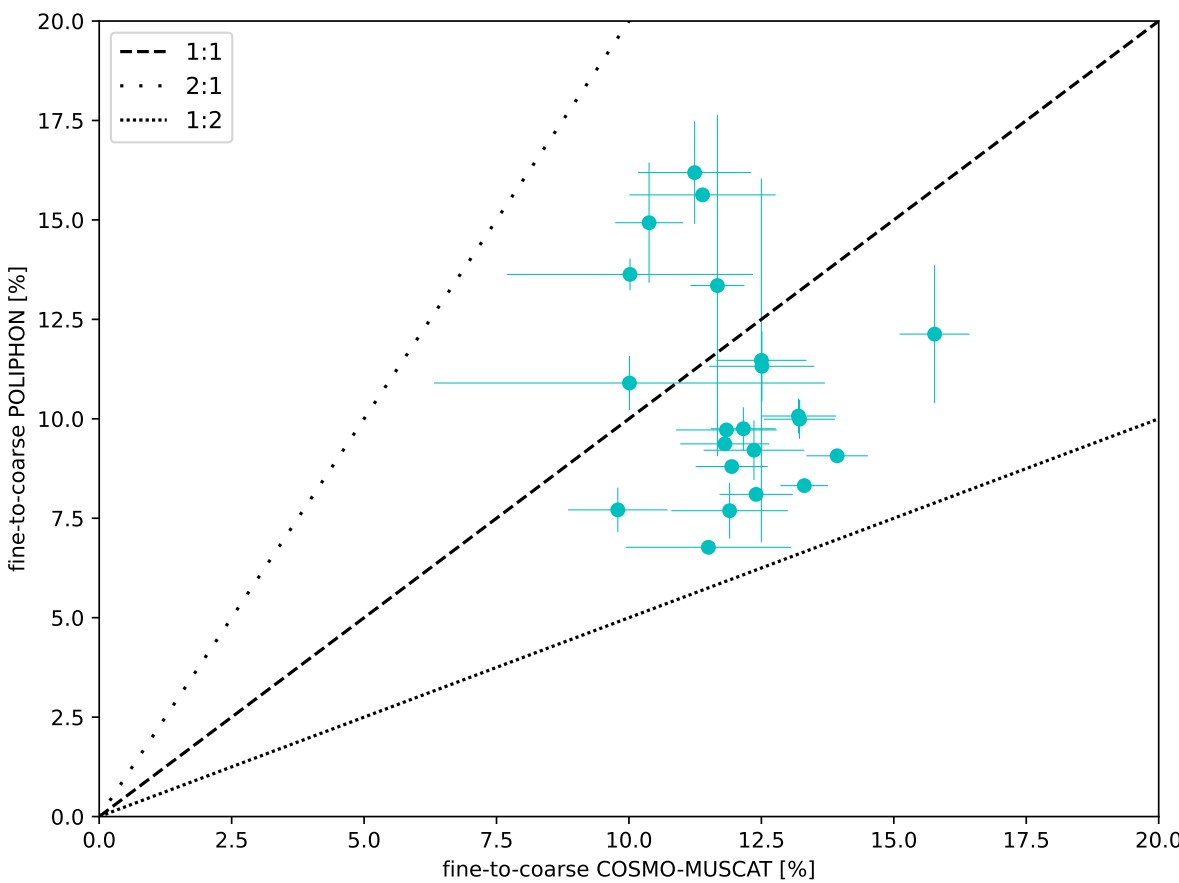

**Figure 6.** Fine-to-coarse dust mass concentrations comparison between COSMO-MUSCAT and POLIPHON. The dashed line represents the 1:1 line and the dotted lines represent the 2:1 and 1:2 lines as indicated in the legend. The error bars represent the standard deviation of the ratio per dust layer.

dust plumes in the region, capturing the general patterns of timing and intensity, particularly for the periods of time when the

$\text{ÅE}(\alpha)_{440/870}$ are below 0.3, indicating that dust is the the main aerosol type.

These findings suggest that the model captures well the dust transport across the region. Furthermore, considering that dust plumes during this season typically travel above the marine boundary layer (Schepanski et al., 2009), the dust observed over Cabo Verde is mostly part of the larger regional transport patterns rather than originating from local emissions. The results are presented and further discussed in the Appendix A.

**3.2.1   Dependence of optical properties on hematite fraction**

As outlined in the introduction, changes in the hematite content of mineral dust increase the imaginary part of dust's complex refractive index (CRI) in the UV-Vis spectral range, dominating light absorption (Di Biagio et al., 2019) as well as decreasing

the backscattering coefficients (Chang et al., 2024; Veselovskii et al., 2016; Wandinger et al., 2023). Assuming constant particle shape and size, we expect that increasing hematite content should lead to a reduction in the particle depolarization ratio, as suggested by the experimental findings of Miffre et al. (2023), and to a reduction in the backscatter coefficients at both 355 nm and 532 nm. Crucially, this decrease is considered to be spectrally dependent, with a steeper decline at 532 nm than at 355 nm, a pattern supported by Mie theory (De Leeuw and Lamberts, 1987) through size parameter considerations: for the same particle radius, the size parameter $\chi = 2\pi r/\lambda$ is larger at 355 nm than at 532 nm, leading to a more forward-peaked scattering phase function at the shorter wavelength. As a result, 355 nm backscattering is initially lower and less influenced by internal and surface resonances. In contrast, 532 nm backscatter benefits more from these resonance enhancements, which are more strongly dampened as the imaginary part of the CRI increases and consequently absorption increases, leading to the sharper decline of the backscattering coefficient at this wavelength.

While we acknowledge that non-sphericity of mineral dust significantly influences its optical properties, we adopt the perspective that the spectral behavior of the backscatter response, namely, its dependency on the imaginary part of the CRI, is preserved across different particle shapes. This viewpoint is supported by the comparative study of Chang et al. (2024), which showed that the shape of the backscatter coefficient's dependency on the imaginary part of the CRI remains consistent across various scattering models, including such as those by Dubovik et al. (2006) and Saito et al. (2021). Notably, Chang et al. (2024) also reports that the backscatter-related Ångström exponent for 355-532 nm increases for larger particles as the imaginary part of the CRI increases, reinforcing the expected wavelength-dependence suppression of the backscatter coefficient. As a result of this differential sensitivity, the lidar ratio ($S = \alpha/\beta$) is expected to increase with hematite content due to the stronger suppression of the backscatter coefficients relative to the extinction coefficients.

Moreover, this spectral imbalance in backscattering manifests in the backscatter-related Ångström exponent for the UV-Vis wavelengths, defined as:

$$\text{ÅE}(\beta)_{355/532} = -\frac{\ln(\beta_{532}/\beta_{355})}{\ln(532/355)}. \tag{1}$$

When $\beta_{532}$ is greater than $\beta_{355}$, $\text{ÅE}(\beta_{355/532})$ is negative. This is the case for all dust plumes analyzed here. It follows, that if both coefficients decrease with an increasing hematite content, particularly with $\beta_{532}$ dropping faster than $\beta_{355}$, their ratio approaches unity, causing $\text{ÅE}(\beta_{355/532})$ to shift towards zero. This behavior is clearly observed in Fig. 7, supporting the hypothesis that hematite fraction influences the spectral shape of the backscattering in a predictable and physically meaningful manner.

In real atmospheric conditions, however, the relationship between hematite content and optical properties is complicated by differences in particle size and shape, which are known to strongly influence lidar-derived intensive optical properties. Figure 7 shows the correlation between the modeled average hematite fraction and lidar-retrieved intensive optical properties across 22 dust-dominated layers. No significant correlation is found between hematite content and lidar ratios or particle depolarization ratios at either 355 nm or 532 nm, suggesting that the effects of hematite on these properties are likely masked by competing influences from size and shape variability. However, a moderately strong positive correlation ($R^2 = 0.49$) is observed between

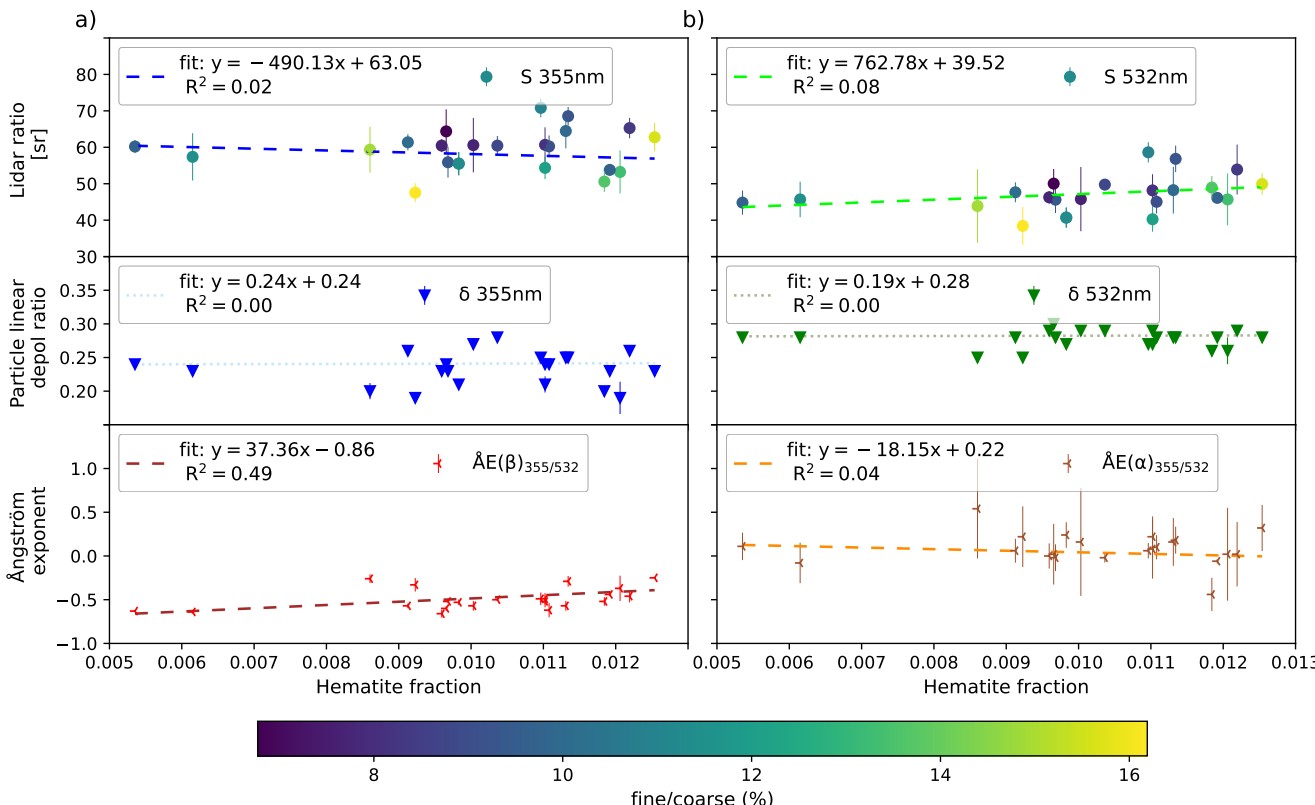

**Figure 7.** Mean values for 22 dust layers, with hematite fractions are shown in the x-axis and three intensive optical properties are shown in the y-axis in each panel. Lidar ratios are shown in color-coded dots, its standard deviation is shown by the error bar. Triangle symbols represent the particle linear depolarization ratios with is standard deviation in the error bar, and the Ångström exponents are depicted by the 45° tilted "Y" symbols. Linear relationships between hematite fraction and each intensive optical property are illustrated by dotted lines. The corresponding linear fit equations and the R-squared value are presented in the legend alongside each variable; the units of the variables in the fit equations are as indicated in their respective axis. (a) illustrates $S_{355}$, $\delta_{355}$, and $\mathring{A}E(\beta)_{355/532}$, (b) shows the values related to $S_{532}$, $\delta_{532}$, and $\mathring{A}E(\alpha)_{355/532}$. The color coding scale of the lidar ratio depends on the fine-to-coarse fraction obtained from the POLIPHON method, values in percentage are shown by the color bar.

$\mathring{A}E(\beta)_{355/532}$ and hematite fraction. This result aligns with our theoretical framework and provides direct observational support for the wavelength-dependent impact of hematite on backscattering efficiency.

As hematite content increases, the backscatter coefficient spectrum flattens, especially due to the stronger attenuation of $\beta_{532}$, leading $\mathring{A}E(\beta)_{355/532}$ to increase. This indicates a reduced spectral gradient in the backscatter signal, which is consistent with a wavelength-dependent absorption increase driven by the imaginary part of the CRI. These findings highlight $\mathring{A}E(\beta)$ as the most sensitive lidar optical parameter to hematite content, reinforcing its potential as a compositional tracer in dust characterization.

The color coding in the lidar ratio analysis in Fig. 7 represents the fine-to-coarse fraction in percentage. Noteworthy, the points with the highest fine-to-coarse fraction (16%), shown in yellow, lie outside of the range of the other $S$ values. Lower fine-to-coarse percentages are clustered around hematite fractions of 0.01 to 0.012. This visual separation suggests that the size distribution may modulate the response of these intensive optical properties to hematite, with fine-mode enrichment possibly enhancing or obscuring compositional signals. Given the known nonlinear dependence of intensive properties on particle size, shape, and composition (Chang et al., 2024; Huang et al., 2023; Miffre et al., 2020, 2023; Saito and Yang, 2021; Wandinger et al., 2023), it is not surprising that only $\text{ÅE}(\beta)$ shows a strong correlation. Prior studies such as Di Biagio et al. (2019) and Adebiyi and Kok (2020) have shown that absorption efficiency increases with particle diameter, while scattering efficiency decreases, further complicating simple linear interpretations.

For this reason, separating dust cases based on fine-to-coarse ratio could improve interpretation, by controlling the variability in particle size distributions between layers. This approach may help clarifying whether the observed trends truly reflect hematite-driven changes on the imaginary part of the CRI.

Furthermore, the lidar ratio related Ångström exponent for the 355 and 532 nm wavelengths ($\text{ÅE}(S)_{355/532}$) exhibits a weak negative correlation with hematite. This suggests that the hematite fraction may influence the lidar ratio with a wavelength dependence, potentially by impacting more $S_{532}$ than $S_{355}$. This influence could be attributed to the effect of hematite on the imaginary part of the CRI through its impact on the backscattering properties. However, with a low R-squared value of 0.25, this relationship remains weak and non-significant to provide any conclusions (see Fig. B1).

While for this study dust-dominated layers were carefully selected, the presence of other aerosols, such as those from anthropogenic activity, e.g., black carbon and biomass burning, cannot be entirely ruled out. These aerosols, particularly black carbon, are known to absorb radiation, specially in UV part of the spectrum (Li et al., 2022; Tesche et al., 2011), which can potentially influence the measured optical properties. However, we minimized the potential impact by applying the $\text{ÅE}(\alpha)_{440/870}$ criterion for the dust layer selection (see step 1 in Fig. 1).

Interestingly, the observed moderately linear relationship between $\text{ÅE}(\beta)_{355/532}$ and hematite fraction is not reflected in the lidar ratio analysis. This is partly explained by the lack of correlation with the extinction-related Ångström exponent ($\text{ÅE}(\alpha)_{355/532}$), thereby diminishing the influence of hematite on dust backscattering properties. Given the methodology used for this study, an intriguing opportunity is presented in the following section in order to explore whether limiting size differences within the dust plumes could potentially strengthen correlations between hematite fraction and intensive optical properties.

### 3.2.2 Size dependency

To gain a clearer understanding of the relationship between the measured intensive optical properties and the modeled hematite fraction, the study cases were separated into three artificial clusters based on different fine-to-coarse dust mass ratio (see Fig. 3). This clustering approach aims to reduce size differences within the dust layers, mitigating the size-dependent effects of the lidar ratio and the backscatter-related ÅE.

Figure 8 presents the relationships between the modeled hematite fraction and measured intensive optical properties, stratified into three clusters based on the fine-to-coarse dust mass ratio. Corresponding mean values and correlation coefficients

**Table 3.** Mean values and the linear correlation coefficient, $R^2$, with respect to changes of the hematite fraction with lidar ratios ($S$), particle linear depolarization ratios ($\delta$), backscatter-related Ångström exponent ($\text{ÅE}(\beta)$) and extinction-related Ångström exponent ($\text{ÅE}(\alpha)$) for the 355 nm and 532 nm wavelengths. Categorized according to size differences throughout the case studies following Fig. 3 based on the fine-to-coarse ratio (f/c) from the data obtained by appying the POLIPHON method.

| fine(f)/coarse(c) | $S_{355}$ [sr] | $\delta_{355}$ | $S_{532}$ [sr] | $\delta_{532}$ | $\text{ÅE}(\beta)_{355/532}$ | $\text{ÅE}(\alpha)_{355/532}$ |
|---|---|---|---|---|---|---|
| f/c > 11% | 55±4 | 0.208±0.009 | 44±5 | 0.265±0.007 | -0.43±0.06 | 0.13±0.31 |
| (8 cases) | $R^2$=0.01 | $R^2$=0 | $R^2$=0.19 | $R^2$=0.05 | $R^2$=0.55 | $R^2$=0.01 |
| 9% < f/c < 11% | 62±3 | 0.244±0.005 | 49±3 | 0.279±0.003 | -0.52±0.04 | 0.07±0.14 |
| (8 cases) | $R^2$=0.10 | $R^2$=0.01 | $R^2$=0.05 | $R^2$=0.03 | $R^2$=0.63 | $R^2$=0.19 |
| f/c < 9% | 62±4 | 0.253±0.004 | 49±5 | 0.292±0.002 | -0.55±0.04 | 0.04±0.31 |
| (6 cases) | $R^2$=0.47 | $R^2$=0.51 | $R^2$=0.04 | $R^2$=0.37 | $R^2$=0.60 | $R^2$=0.02 |

for each group are provided in Table 3. Panels (a-b) show the cases with the highest fine-to-coarse ratios, reflecting larger proportions of fine and, consequently, the greatest particle size differences within a dust layer. In contrast, panels (e-f) display the cases with the largest coarse dust portion, indicating less drastic size differences within each dust layer. The intermediate group is shown in panels (c-d).

The validity of the size-based clustering is further supported by the trends in the mean values of the intensive optical properties across the groups, as shown in Table 3. As expected from theoretical and observational studies, the lidar ratio ($S$) increases with particle size due to the greater contribution of coarse particles to extinction relative to backscatter. This trend is clearly reflected in the mean $S$ values across the clusters, confirming that the size separation effectively captures systematic size-dependent trend in the optical responses. Similarly, the particle linear depolarization ratio ($\delta$) increases with particle size, consistent with findings by Hofer et al. (2020) and Miffre et al. (2023), who demonstrated that larger and more irregularly shaped particles produce higher depolarization. In our data, mean $\delta_{355}$ values increase from 0.21 in the fine-mode cluster (Fig.8a) to 0.25 in the coarse-mode cluster (Fig.8e), while a similar rise is seen in $\delta_{532}$.

However, particle depolarization ratios, remain largely uncorrelated with hematite fraction among size clusters. This is consisting with prior findings, given that depolarization is primarily sensitive to particle shape and less so to composition. While the laboratory study by Miffre et al. (2023) reported that the presence of hematite tends to lower $\delta_{355}$, this is not reflected in the current dataset. A weak negative trend is only hinted at in the coarse-mode cluster (Fig. 8f), where a modest negative correlation is observed between $\delta_{532}$ and hematite fraction is observed for the coarser particles case. However, given the limited number of samples in this cluster, the result should be interpreted cautiously.

By contrast, the backscatter-related Ångström exponent ($\text{ÅE}(\beta_{355/532})$) consistently exhibits moderate and statistically meaningful correlations with hematite fraction in all size clusters. The strongest correlation is observed in the intermediate size group (Fig. 8c, $R^2$=0.63), followed closely by the coarse mode group (Fig. 8e, $R^2$=0.60). The fine-mode group also shows a notable, though slightly weaker, correlation (Fig. 8a, $R^2$=0.55). Notably, the correlation coefficient, $R^2$, increases positively when moving from cases with the higher fine dust portion to those with lower proportions, as seen in Fig. 8(a,b) compared to

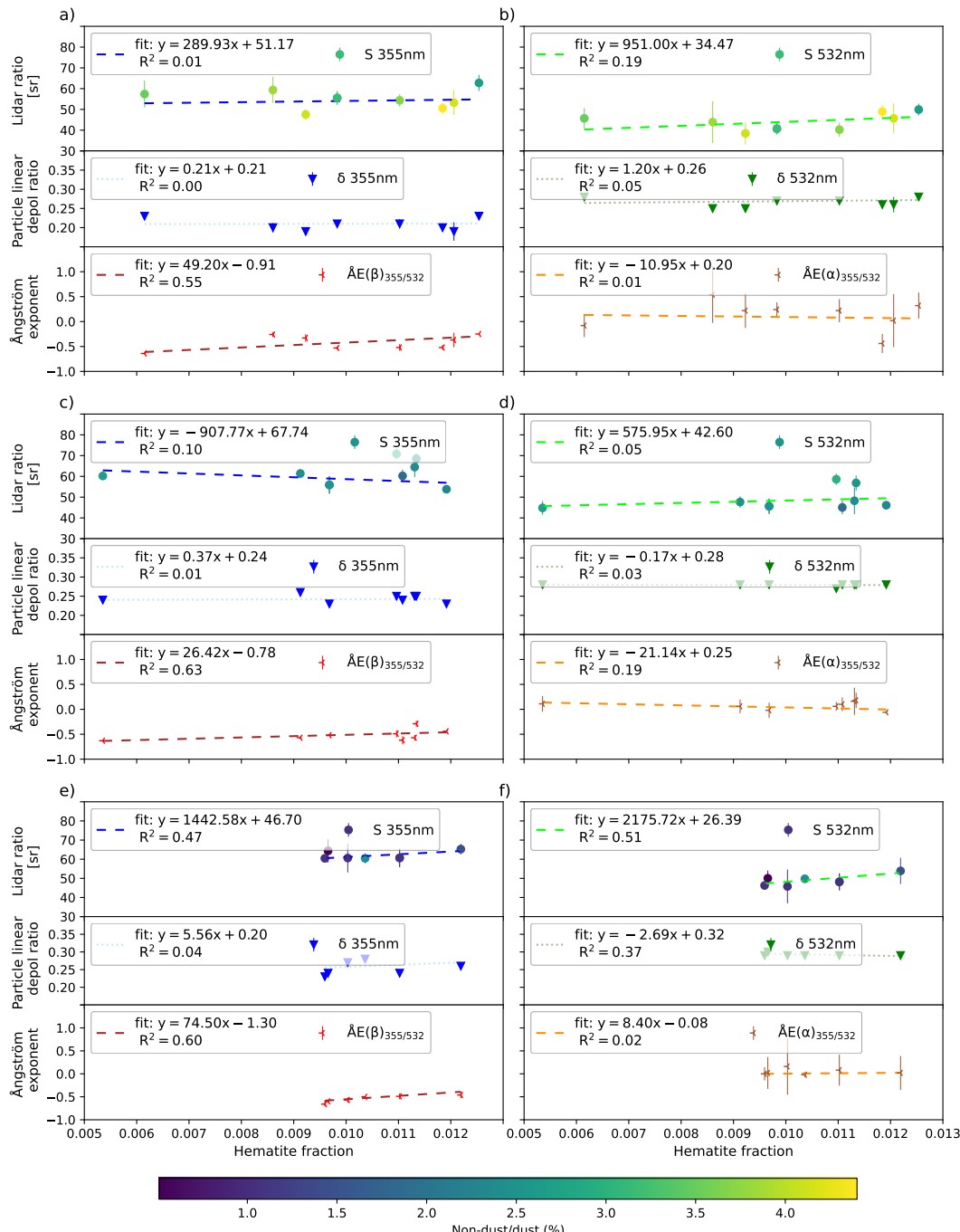

**Figure 8.** As in Fig. 7 but the color coding differs and the twenty two dust cases are separated regarding their fine-to-coarse fraction: (a-b) dust layers with fine/coarse above 0.11, (c-d) dust layers with fine/coarse above 0.09 and below 0.11, and (e-f) dust layers with fine/coarse below 0.09. (a,c,e) illustrate $S_{355}$, $\delta_{355}$, and $\mathring{A}E(\beta)_{355/532}$, (b,d,f) shows the values related to $S_{532}$, $\delta_{532}$, and $\mathring{A}E(\alpha)_{355/532}$. The color coding scale depend on the non-dust to dust fraction obtained from the POLIPHON method, values in percentage are shown by the color bar.

Fig. 8(e,f), suggesting a stronger relationship between hematite fraction and backscattering properties in coarser dust layers. The consistently positive slopes of the linear fits across all size clusters further aligns with theoretical considerations that as hematite content increases, and with it, the imaginary part of the CRI, the backscatter coefficient at 532 nm decreases more steeply than at 355 nm, leading to a convergence of the two values and thus driving $\text{ÅE}(\beta_{355/532})$ toward zero.

Conversely, the extinction-related Ångström exponent ($\text{ÅE}(\alpha)_{355/532}$) shows no meaningful correlation with hematite frac-
545 tion across the size clusters (Fig. 8b,d,e), nor any consistent trend. However, the average values of $\text{ÅE}(\alpha)_{355/532}$ decrease towards zero with increasing coarse-mode dominance. This trend highlights the strong sensitivity of $\text{ÅE}(\alpha)$ to particle size, and suggests that unlike $\text{ÅE}(\beta)$, it is almost decoupled from the change of the imaginary part of the CRI to larger values from Vis to UV.

The relationship between hematite fraction and lidar ratio shows a gradual strengthening across size clusters. For $S_{355}$, the
550 correlation shifts from non-significant in the fine-mode group to moderately positive in the coarse-mode group (Fig. 8a,e). A similar trend is observed for $S_{532}$ (Fig. 8b,f). Although the sample size is small, particularly in cases with larger proportion of coarse particles (six samples), the progressively stronger correlation coefficients, combined with the well-established influence of changes in the imaginary part of the CRI on backscattering properties, suggests a potentially meaningful relationship between the hematite fraction and the lidar ratio, with a more marked tendency for the Vis portion of the spectrum at 532 nm (Chang
et al., 2024; De Leeuw and Lamberts, 1987). This is consistent with theoretical predictions, as the backscatter coefficient at 532 nm is more strongly suppressed by increases in the imaginary part of the CRI than at 355 nm. The relatively higher sensitivity of $S_{532}$ compared to $S_{355}$ may reflect both spectral dependencies in the imaginary part of the CRI and the intrinsic wavelength-dependent response of the backscatter coefficient. This finding align with previous studies on the spectral impact of hematite fraction on absorption. For example, Di Biagio et al. (2019) found a higher $R^2$ when analyzing the changes of the
hematite fraction relative to the single scattering albedo at 520 nm ($R^2$=0.78) compared to 370 nm ($R^2$=0.73).

This analysis highlights that segregating dust plumes into size-based clusters, which reduce size variability within the ana-lyzed dust plumes, enhances the strength and clarity of the correlations between modeled hematite fractions and optical prop-erties. This most consistent improvement is observed for the backscatter-related ÅE across the UV-Vis spectral range, with moderately positive correlations emerging across all three size classes in Fig. 8. This consistency suggest that $\text{ÅE}(\beta)_{355/532}$
is particularly sensitive to compositional diversity, especially when they affect the imaginary part of the CRI. In contrast, this trend is less uniform across other lidar-derived intensive optical properties, reflecting the complex interplay between particle size, shape, and composition. A plausible explanation is that coarser particles exhibit enhanced absorption relative to scattering, as shown in previous studies (Adebiyi and Kok, 2020; Di Biagio et al., 2019), which would amplify the impact of hematite content on optical parameters like the lidar ratio and $\text{ÅE}(\beta)$. This is consistent with the stronger correlations observed for $S$ in
the coarser particle cluster.

Following Miffre et al. (2020); Schuster et al. (2012); Veselovskii et al. (2020) and Wandinger et al. (2023), it is well estab-lished that the backscatter-related ÅE for the 355 to 532 nm wavelengths is influenced by the imaginary part of the CRI. Even though the negative relationship found in Veselovskii et al. (2020)'s sensitivity study, where $\text{ÅE}(\beta)_{355/532}$ decreases as the imaginary part of the CRI increases, does not align with our findings, Miffre et al. (2020)'s laboratory study suggest a different

scenario. In cases with larger effective radius and considering non-spherical particles, where the real part of the CRI does not vary, $\mathrm{\AA{E}}(\beta)_{355/532}$ increases towards zero as the imaginary part of the CRI increases. A reasonable interpretation of this discrepancy lies in the design of the Veselovskii et al. (2020) simulations, where only the imaginary part of the CRI at 355 nm was varied, while keeping it fixed at 532 nm. As a result, their study captures only the decrease in the backscatter at 355 nm, leading to a reduction in $\mathrm{\AA{E}}(\beta)_{355/532}$. This approach inherently overlooks the wavelength-dependent sensitivity of backscattering to absorption, particularly the stronger suppression of $\beta_{532}$ with increasing imaginary part of the CRI. Consequently, their results do not capture the behavior observed in our data (Fig. 7 and Fig. 8)

In our study, we consider that hematite content is the main contributor to the variability in the imaginary part of the CRI. This is supported by Di Biagio et al. (2019), who observed negligible changes in the real part of the CRI across various dust samples, while showing a strong positive correlation between hematite content and the imaginary part, particularly in the UV-Vis range (see Fig.9 in their study). Therefore, the observed increase in $\mathrm{\AA{E}}(\beta)_{355/532}$ with hematite fraction in our results is best explained by hematite-driven increases in the imaginary part of the CRI, which lead to wavelength-dependent suppression of backscattering, especially at 532 nm. This interpretation is further supported by the fact that the strongest $\mathrm{\AA{E}}(\beta)_{355/532}$ correlations appear in the size clusters where particle size variability is minimized, allowing the impact of hematite increasing to become more apparent.

The lidar ratio-related $\mathrm{\AA{E}}$ analysis shows a similar results to $\mathrm{\AA{E}}(S)_{355/532}$, showing the strongest correlation with the varying hematite fraction for the intermediate cluster ($R^2$=0.42). Furthermore, $\mathrm{\AA{E}}(S)_{355/532}$ correlation coefficients get stronger between the cases with finer particles and the ones with coarser particles (see Fig. B2). These results reinforce the idea that hematite influences the lidar ratio with a wavelength dependence by impacting stronger $S_{532}$ than $S_{355}$.

Additionally, the color coding provides further insight by highlighting the relationship between the non-dust to dust fraction, as derived from POLIPHON, within each dust layer. As shown in Fig. 8, the highest non-dust portions are found in the layers with the larger fine dust portion Fig. 8(a-b), and this ratio decreases progressively as the fine dust portion diminishes. The presence of higher non-dust content, such as soot, whether locally produced or transported within the layer, can influence the measured optical properties by absorbing radiation at UV-Vis wavelengths (Müller et al., 2009). This effect contributes into the understanding of the lack of clear, positive relationships between the $S_{355}$, $S_{532}$, and the increase of hematite content for cases a higher portion of non-dust aerosols.

We conclude that the complex interactions between particle size, shape and composition with the intensive optical properties make the effect of increased hematite fraction more apparent only when dust cases are size segregated. This approach reduces size differences and minimizes interference from other aerosols. Although these separations are artificial due to the uncertainties of the fine-to-coarse ratios, they serve as a valuable exploratory analysis, highlighting the importance of size segregation in studying the impacts of iron oxide content on intensive optical properties.

## 4 Conclusion and implications

Positive correlations are revealed in the analysis between lidar retrieved intensive optical properties and modeled hematite fraction in dust cases when considering particle size separately, especially for layers with minimal fine dust and non-dust content. This suggests that both particle size and composition significantly influence dust optical properties, although their effects are intertwined and difficult to disentangle due to their nonlinear nature. Furthermore, due to the nature of the POLIPHON method, the concentration of non-dust particles is contained within the fine fraction therefore obscuring the correlation between hematite and optical properties in this size range.

Regarding the positive correlation found with the backscatter-related Ångström exponent, our findings align with Miffre et al. (2020), indicating that an increase in the imaginary part of the CRI, linked to higher hematite content, can lead to an increase in $\text{ÅE}(\beta)_{355/532}$, particularly when intial values are negative. This supports the notion that changes in $\text{ÅE}(\beta)_{355/532}$ are primarily driven by differences in the imaginary part of the CRI, rather than the real part. This is consistent with previous studies, such as those by Di Biagio et al. (2019), which found no change in the real part of the CRI but identified hematite's impact on the imaginary part. These correlations indicate a negative relationship between the hematite fraction and the dust's backscattering properties, with a wavelength-dependent influence. While the lidar ratio does not show a strong correlation, a clear trend emerges as particle size differences become more constrained at both 355 nm and 532 nm wavelengths. In the size segregated cases, the positive correlation strengthens. This trend is further supported by the lidar ratio-related ÅE correlation analysis.

This study presents a framework for understanding the influence of hematite content on lidar measurements. To enhance the statistical validity of these findings, it is essential to identify additional lidar retrieved dust plumes that meet this study criteria, potentially using machine learning techniques. Expanding this research to other desert regions globally is also recommended, given the lidar ratio variability across different deserts (Hofer et al., 2020; Schuster et al., 2012).

Future modeling efforts should include a parametrization that takes into account the changes in particle size distribution as minerals get emitted, as well as adding the soil distribution content of goethite. Additionally, a new development could replace the use of soil interpolation produced databases by providing a mineralogical dataset from spaceborne hysperspectral measurements through NASA's Earth Surface Mineral Dust Source Investigation (EMIT: https://earth.jpl.nasa.gov/emit/).

Moreover, if a specific combination of lidar retrieved measurements can be identified for dust layers with varying hematite content, these measurements could be linked to distinct dust source regions with unique mineralogical distributions (Formenti et al., 2014; Go et al., 2022). This would build on previous studies that have linked differences in measured lidar ratios to dust originating from different regions of the Sahara Desert (Esselborn et al., 2009). To advance this research, efforts could also incorporate the infrared channel from Polly[XT] retrievals to identify minerals interacting with this part of the spectrum (Gebauer et al., 2024; Haarig et al., 2022, 2025).

To conclude, although a positive correlation has been identified between the backscatter-related Ångström exponent and hematite fraction, it is clear that further research is necessary in order to materialize and characterize this relationship and further relationships between hematite content and lidar retrieved intensive optical properties, specially considering the non-

640 sphericity of the particles. Nonetheless, these findings underscore the importance of considering both size and compositional factors for accurately representing dust optical properties. This could help reduce uncertainties in model estimates of dust and its direct radiative effect.

*Data availability.* The dataset for reproducing the graphs presented here are available at http://doi.org/10.5281/zenodo.13908845.

Raw polly lidar observations (level 0 data, measured signals) can be accessed through the PollyNet database (http://polly.tropos.de/, last access on February 2025).

**Appendix A: Comparison between COSMO-MUSCAT-derived and AERONET AOT**

A comparison of aerosol optical thickness (AOT) at 550 nm was conducted using AERONET data to evaluate the model's performance. The model derived AOT is calculated based on the simulated total column dust mass loads, an effective radius, a dimensionless dust extinction efficiency at 550 nm, which vary within the model size classifications, and a fixed, homogeneous density of 2650 kg/m$^3$. Further details of this diagnostic approach can be found in Gómez Maqueo Anaya et al. (2024). The comparison covered the entire simulation periods, from 8 August to 30 September 2021 and from 2 June to 31 July 2022. AERONET stations were strategically selected along the dust transport pathway from the Sahara Desert towards the Atlantic, and level 2.0 data, which undergo cloud screening, were used. The results are depicted in Fig. A1 for 2021 and Fig. A2 for 2022.

COSMO-MUSCAT performance varies across stations. While the model tends to underestimate AOT at certain stations, such as Mindeo and Banizoumbou during JATAC 2021, and Santa Cruz in both campaign periods, it generally overestimates AOT at the other studied stations. Among the stations analyzed, Santa Cruz station showed the best overall agreement with the modeled values. However, the lack of measurements at some stations during significant periods of time in both campaigns introduces bias into the model-observation correlations. Specific events, such as on 9 August 2021 at Cinzana and the midday of 23 August 2021 at Mindelo are significantly over represented and under represented by the model, respectively. The dust event recorded at the Banizoumbou station on the 13 September 2021 also appears to be under represented. However, satellite observations suggest potential cloud contamination in AERONET AOT values despite cloud screening procedures.

It is important to highlight that AERONET AOT retrievals account for all aerosol types potentially present, whereas the model simulates only dust aerosols. To ensure a more direct comparison, the focus is placed on periods when $\text{ÅE}(\alpha)_{440/870}$ values are below 0.3, which serves as a reliable indicator that dust is the dominant aerosol type (Ansmann et al., 2019). Despite these challenges, the model effectively reproduces the temporal evolution of dust plumes in the region, successfully captures the general patterns in timing and intensity. This suggest that the model provides a reliable representation of dust transport across the region and can be confidently used to simulate the life cycle of mineral dust aerosols.

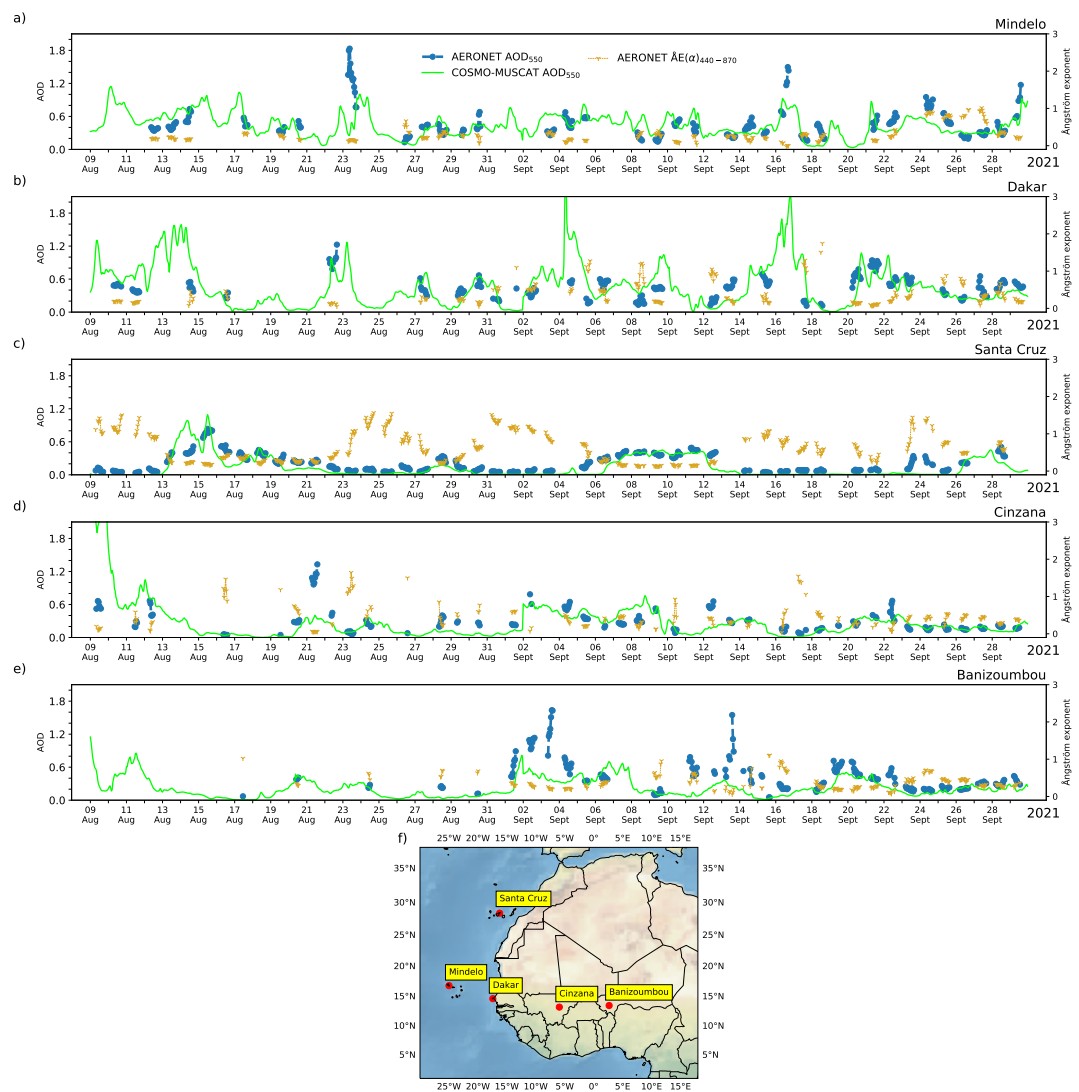

**Figure A1.** Dust AOT at 550 nm calculated from COSMO–MUSCAT dust concentration fields (green) and AOT at 550 nm and the exctintion-related Ångström exponent for the 440–870 nm wavelength range ($\text{ÅE}(\alpha)_{440/870}$) from AERONET, level 2.0, sun-photometer measurements (blue and gold), where each x-axis tick represents 12:00UTC for each day in the range of 8 August to 30 September 2021. Five different stations across the Sahara and downwind locations are shown. a) Mindelo (16.878°N, 24.995°W; Cape Verde), b) Dakar Belair (14.702°N, 17.426°W; Senegal), c) Santa Cruz Tenerife (28.473°N, 16.247°W; Spain), d) IER Cinzana (13.278°N, 5.934°W; Mali), e) Banizoumbou (13.547°N, 2.665°E; Niger), (f) AERONET station geographic locations and names used in this comparison.

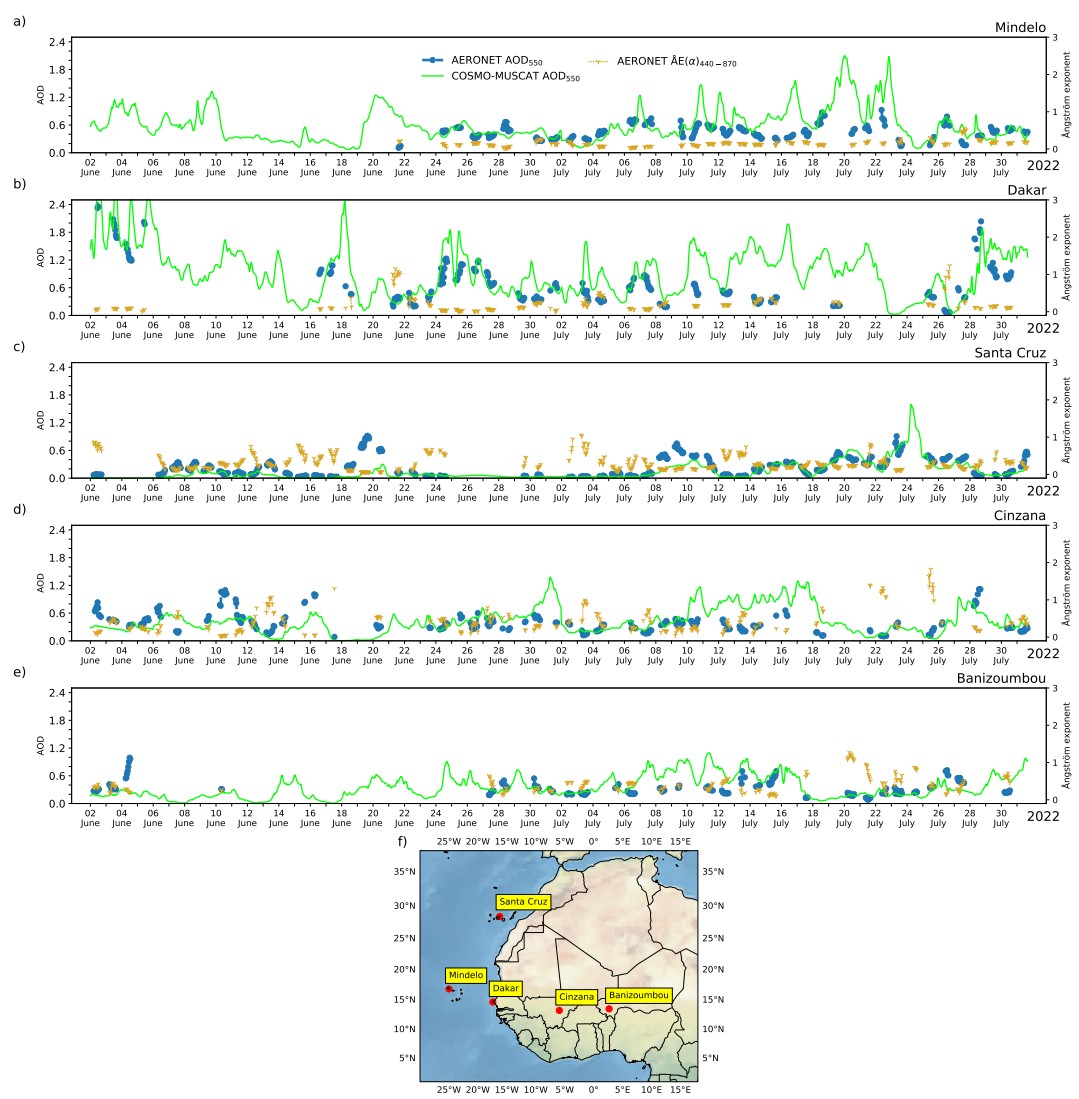

**Figure A2.** As in Fig. A1 but for the period of 2 June to 31 July 2022.

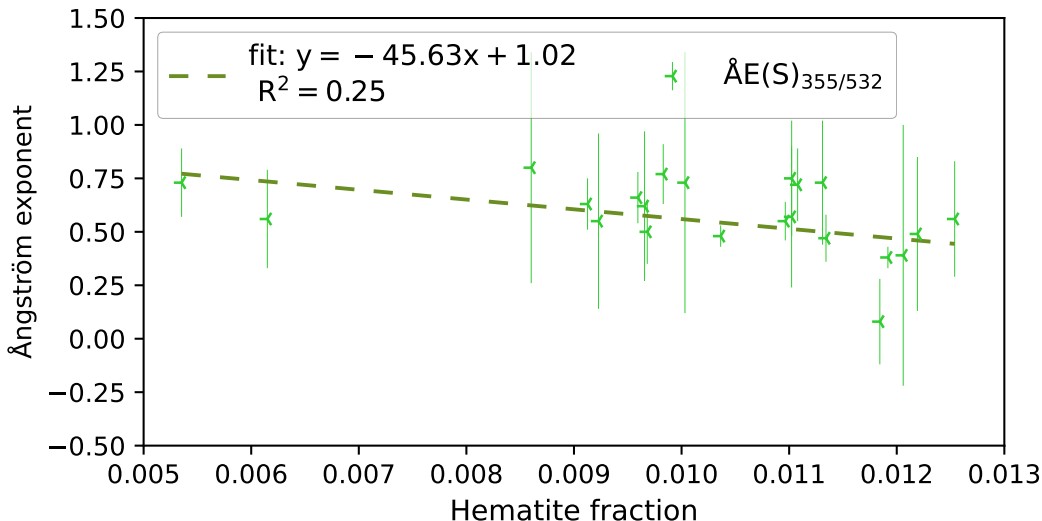

**Figure B1.** Mean values for 22 dust layers, hematite fractions are shown in the x axis and the lidar ratio-related Ångström exponent for the 355 and 532 nm is shown in the y axes. The standard deviation of $\mathring{A}E(S)_{355/532}$ is shown by the error bar. Linear fit between the hematite fraction and $\mathring{A}E(S)_{355/532}$ is illustrated by the dotted lines, the R squared value of the linear fit is shown in the dashed line legend.

### Appendix B: Lidar ratio Ångström exponent vs hematite fraction

Figures B1 and B2 show the correlation analysis between averaged modeled hematite fraction and the lidar ratio-related Ångström exponent considering 22 dust layers. The lidar ratio-related Ångström exponent is derived from lidar measurements in the following way:

$$
\begin{aligned}
\mathring{A}E(S)_{355/532} &= \frac{\ln(S_{355}/S_{532})}{\ln(355/532)} \\
&= \frac{\ln(\alpha_{355}/\beta_{355} * \beta_{532}/\alpha_{532})}{ln(355/532)} \\
&= \frac{\ln(\alpha_{355}/\alpha_{532}) + \ln(\beta_{532}/\beta_{355})}{ln(355/532)} \\
&= \mathring{A}E(\alpha)_{355/532} - \mathring{A}E(\beta)_{355/532}
\end{aligned} \tag{B1}
$$

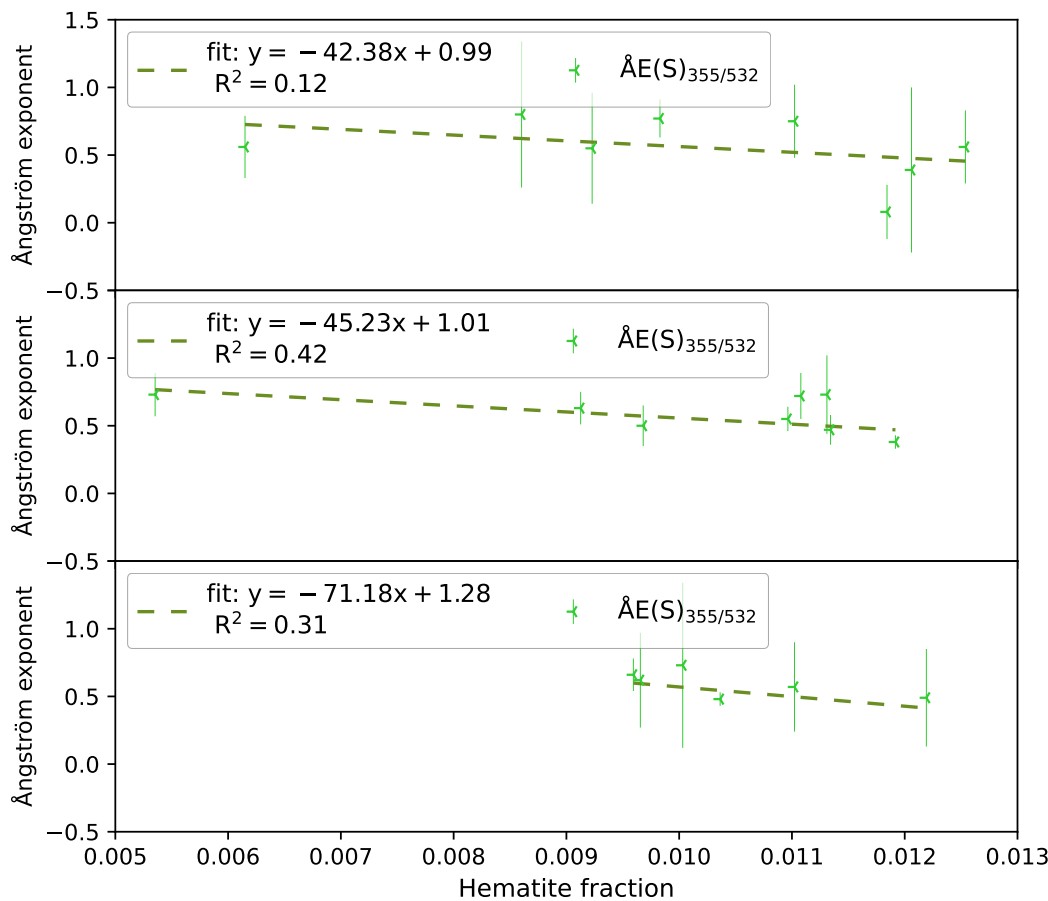

**Figure B2.** Same as Fig. B1 but for the case studies being separated regarding their fine-to-coarse fraction: first row shows dust layers with fine/coarse above 0.11, middle row illustrates dust layers with fine/coarse above 0.09 and below 0.11, and last row depicts dust layers with fine/coarse below 0.09.

*Author contributions.*  SGMA wrote the manuscript draft; DA, KS, MH, JH, and BeHe reviewed and edited the manuscript; DA, KS, HB, and AA provided resources such as study materials, instrumentation, and analysis tools and were part of the conceptualization of the project; SGMA, DA, UW, MH, JH, HB, and AA were part of the formal analysis of lidar data. SGMA, KS, and DA performed the analysis of lidar data compared to modeled results. MF – software development (restructured the code around MUSCAT dust emission scheme); SGMA, MF, BeHe, KS, and IT contributed to code development surrounding the mineralogy inclusion; AS, BiHe, HB, and RE are part of the maintenance and continuous improvement of the PollyXT lidar device(s).

*Competing interests.*  The authors declare that they have no conflict of interest

*Acknowledgements.*  This study is done in the framework of the DUSTRISK (a risk index for health effects of mineral dust and associated microbes) project, funded by the Leibniz Collaborative Excellence Programme Project (grant number K255/2019).

This research has been supported by the German Federal Ministry for Economic Affairs and Energy (BMWi) (grant no. 50EE1721C). Furthermore, we also acknowledge the support through ACTRIS-2 under grant agreement no. 654109 from the European Union's Horizon 2020 research and innovation programme and ACTRIS PPP under the Horizon 2020 – Research and Innovation Framework Programme, H2020-INFRADEV- 2016-2017, Grant Agreement number: 7395302.

We acknowledge and thank the team of OSCM / INMG for their crucial and ongoing support. We further thank ESA and the ASKOS/-JATAC teams for the organization of the JATAC(s) campaign(s) and their continuous support.

Further thanks are due to the Deutscher Wetterdienst (DWD) for cooperation and support, and to all PIs of the AERONET stations used in this study for maintaining the instruments, obtaining the measurements and providing data.

We want to thank all the TROPOS team involved in the PollyNET, the network dedicated to provide continuous aerosol data from automated Raman-polarizations lidars (Baars et al., 2016) https://polly.tropos.de/.

Muñoz Sabater, J., (2019, 2021) was downloaded from the Copernicus Climate Change Service (C3S) Climate Data Store. The results contain modified Copernicus Climate Change Service information 2020 and 2022. Neither the European Commission nor ECMWF is responsible for any use that may be made of the Copernicus information or data it contains.

Furthermore, ChatGPT was utilized to rephrase and shorten sentences, as well to identify the appropriate prepositions.

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
