# Peer review of "Investigating the link between mineral dust hematite content and intensive optical properties by means of lidar measurements and aerosol modelling"

_EGUsphere, 2024_

## Author Response (AR3)

**Replies to the comments from Anonymous referee #1:**

We would like to start the reply by thanking the referee for reviewing our manuscript. We thank you for sharing your insights, thoughts and opinions and we regret that you feel disappointed with our manuscript. We attempt to address your concerns. We have copied the comments into this document; the referee's comments are in Times New Roman blue font while our answers are in Calibri black font. Line numbers refer to the version of the manuscript with track changes. The main body of changes done in order to address the referee's comments can be found in the introduction section between L39-69, the whole section 2.5 regarding the POLIPHON method, and throughout the results sections, in section 3.2.1 and 3.2.2.

This paper is an attempt to relate several dust optical properties (lidar ratios, particle depolarization ratios, backscatter Angstrom Exponents, and extinction Angstrom Exponents) to the fraction of hematite in mineral aerosols. The authors use lidar measurements for the optical properties and the COSMO-MUSCAT model for the hematite fractions. Since the hematite fractions are not measured and modeling the mineralogy of aeolian dust is presently not robust, it is unsurprising that the correlations between the measurements and the model are rather poor.

I am not sure why the authors think that the mass fraction of hematite has an effect on the scattering field of atmospheric dust, even if they had measured the mineralogy (instead of modeled it). Hematite fractions are quite low in mineral aerosols, generally less than ~5% by mass. Thus, varying hematite from its minimal mass fraction (0%) to its maximal mass fraction (~5%) does not alter the real refractive index or the scattering field significantly. And indeed, that is how it turned out -- the correlations presented here are quite poor.

We agree with the reviewer that hematite fractions vary in the order of about 5% (Formenti et al., 2014). That variation has been documented with changes in the absorption capacity of dust by modifying the imaginary part of the complex refractive index (CRI) (Alfaro et al., 2004; Balkanski et al., 2007; Kandler et al., 2009; Lafon et al., 2004, 2006; Sokolik and Toon, 1999; Wagner et al., 2012). Following these published works, no change in the real part of the CRI is measured, while on the other hand, Schuster et al. (2012) points to variations in the real part of the CRI from AERONET dust measurements. However, the optical properties measured with lidar, especially the lidar ratio, depend on both, the real and imaginary part, next to other parameters like size distribution and shape, and thus a change in the optical properties due to changes in absorption capacities is generally plausible. With regards to our manuscript, we don't perform absorption calculations in our studies and we do not suggest a variation in the real part of the CRI. We bring together lidar measurements and modelling results regarding hematite. We make this now clearer in the introduction as stated below.

As stated above, published works suggest that the backscattering properties, and thus also the lidar ratio, of dust change with changes in the imaginary part of the CRI (De Leeuw and Lamberts, 1987). This effect has been measured in the laboratory by Miffre et al. (2020) for the variations of the imaginary part of the complex part of the CRI as measured by DiBiagio et al. (2019). Hence, we are following the literature on the effect of the hematite fraction on

the optical properties measured by a state-of-the-art aerosol lidar. We have added that information in the introduction to clarify our approach; see further L38-44.

On the other hand, hematite and the other iron oxides are responsible for nearly all of the absorption in mineral dust, so one would expect some significant variability in the single-scatter albedo (SSA) or mass absorption efficiency (MAE) associated with the mass fraction of hematite. Unfortunately, the SSA and MAE are not discussed in this paper (probably because they are not available from lidar measurements).

We agree, SSA and MAE are not directly available from lidar measurements, which is stated in Section 2.2, line 217. Furthermore, the influence of changes in hematite on these quantities have been measured, and documented before. For references see measurements by DiBiagio et al. (2019) and recently in the modelling study by Zhang et al. (2024).

The POLIPHON method is layed out in 3 steps, but it lacks details and requires much hand waving. A brief review needs to be provided with enough details for the reader to understand what the authors are doing and the associated uncertainties. For instance, in Step 1 (line 254) the authors say "The initial step involves separating the particle backscatter coefficient based on the particle linear depolarization ratio." How? What depolarization thresholds are you using to separate fine and coarse (if that is how you are doing it), and how do you separate fine dust from fine non-dust? In step 2, lidar ratios are estimated based upon their probable origins (how determined? a model?). Step 3 is an extinction to mass conversion that is based upon AERONET climatologies, but again, no reported conversion factors nor any details of how they were obtained. Then the authors state that an "advantages of this method is that it does not require a dust particle shape model in the data analysis since it relies solely on the measured optical properties." This is not true, though, since AERONET needs to use the spheroid optical model and a density assumption to relate aerosol mass to extinction.

Many thanks for bringing this to our attention. We have added more details regarding the coefficients used for the POLIPHON method and clarifying from where these numbers come from We added the used coefficients to ensure replicability, so, thank you for pointing it out that they are missing. That AERONET uses the spheroid optical model in order to calculate volume mass fractions is true and we have acknowledged it so. The body of modifications done to this section can be found in section 2.5 between L292-340 plus Table 1 now contains all the coefficients used for applying the POLIPHON method.

There are way too many references to figures in other work in this paper (twice on line 52, lines 125, 272, twice on line 281, lines 423, 426). This is laziness, in my opinion, and as a reader I do not want to search through a bunch of extra journal articles to read a paper -- a paper needs to stand on its own merits.

We are sorry to hear that our preciseness in referencing bothers you and would like to explain why we decided to provide such detailed references. We decided to point the reader directly to the figures published in the framework of the referenced works in order to allow for direct access to the referenced results rather than only providing the references in a general way. Those figures are used as reference and they are not needed to understand our manuscript, but some reader may find it helpful to be directed in a precise way. We disagree that the manuscript does not stand in its own merits.

To sum up, the authors don't provide any physical basis to explain why the hematite mass fraction should be related to the extinction, linear depolarization ratio, or Angstrom exponents. So this paper is really an epidemiology study. However, the optical parameters that the authors chose are not physically related to iron oxide content (and the authors have not tried to demonstrate this), so the epidemiology study did not show any skill between the parameters (as expected). Thus, this paper is not suitable for publication, in my opinion.

If the authors truly want to demonstrate a relationship between iron oxide fractions and aerosol optical properties, they need to

1.) replace the model in this paper with measurements, and

2.) choose dust optical properties that are sensitive to absorption (e.g., SSA).

We revised the manuscript to make it clearer and provide a physical basis why the hematite mass fraction is affecting dust backscattering properties, showing up in the lidar ratio, particle depolarization ratio and backscatter-related, lidar ratio-related Angstrom exponents. Furthermore, we state why these properties might be related to iron oxide content. We apologize that this was not clearly communicated in the first version of the manuscript. We hope we were able to improve on the clarity of the presentation of our study.

Line items:

Line 69, Authors state:

"Given that previous studies demonstrated the impact of iron oxides on the extinction (absorption plus scattering) properties, particularly in the UV-VIS specturm, this project hypothesizes that this effect will manifest in the lidar measured intensive optical properties at 355 nm and 532 nm wavelengths."

What previous studies? If true, these need to be cited. Personally, I have never seen papers that link hematite content to extinction. Hematite is typically less than 5% of the total dust mass, so I would be surprised if the variation of hematite from 0-5% has a significant effect on the extinction.

We agree that this sentence may be confusing, and hence we have removed it and stated clearly that our aim for the project is to investigate the impact that the modifications of the complex refractive index due to variations in the hematite content have on the backscattering properties of dust by means of the lidar ratio and the backscatter-related Ångström exponent (L82-88).

Line 83, Authors state "Gonçalves Ageitos et al. (2023) found that GMINER dataset fairly reproduces iron oxide content for the Sahara Desert." Looking at Fig 11 in Goncalves (2023), I have to disagree.

Thank you for pointing this out. We wanted to refer to the hematite mass content instead of the iron oxide mass ratio. We have changed the sentence accordingly and provide the exact

panel of the figure we are referring to in L102. Furthermore, the authors discussed the representativeness of the mineralogical dataset for the region with more detail in their Section 5.3.3.

Lines 275-280: A rather strange partitioning of fine-to-coarse dust mass concentrations is described:
fine/coarse > 11%
9% < fine/coarse < 11%
fine/coarse < 9%

Can the authors demonstrate that their fine/coarse partitioning is accurate enough that it makes sense to define a range of only 2% for the 'intermediate' fine/coarse dust mass concentrations?

We have added a couple of disclaimers throughout the manuscript, pointing out that even though the 2% range is not inside the uncertainty range, we make that division in an exploratory effort in order to see if we could isolate the effect of the hematite variation on the intensive optical properties by constraining size differences between the dust particles found in the layer. Those sentences can be found in Section 2.5, L341-357 and in Section 3.2.2, L593. Thank you for pointing out that clarity was missing.

Figure 7: Figure 7 showes the correlation coefficient (r) and presumably the coefficient of determination ($R^2$) for each panel. How come $r^2$ is not equal to $R^2$?

Thank you for pointing out this discrepancy. Reviewing the results, we have found a typo in the Pearson correlation coefficient and p values calculations that resulted on correlation coefficients that are not weighted, while the R-squared calculation is a result of a weighted linear regression analysis. As a result, we have opted for just using the R-squared parameter to analyze the results. Modifications can be seen in both Figs.7 and 8, Table 3, and throughout the result sections 3.2.1 and 3.2.2.

Line 396-398, Authors state: "Despite thesmall sample size, particularly in cases with larger proportion of coarse particles (just six samples), the strong correlation coefficient suggests a meaningful relationship between the hematite fraction and lidar ratio, at least so for the VIS portion of the spectrum at 532 nm."

A strong correlation coefficent does NOT necessarily suggest a meaningful relationship when the sample size is small.

We have changed the wording regarding the analysis of those results. The changes are found in Section 3.2.2, L528-537.

**References:**

Alfaro, S. C., S. Lafon, J. L. Rajot, P. Formenti, A. Gaudichet, and M. Maille´: Iron oxides and light absorption by pure desert dust: An experimental study, J. Geophys. Res., 109, D08208, doi:10.1029/2003JD004374, 2004.

Balkanski, Y., Schulz, M., Claquin, T., and Guibert, S.: Reevaluation of Mineral aerosol radiative forcings suggests a better agreement with satellite and AERONET data, Atmos. Chem. Phys., 7, 81–95, https://doi.org/10.5194/acp-7-81-2007, 2007.

De Leeuw, G. and Lamberts, C.: Influence of refractive index and particle size interval on mie calculated backscatter and extinction, Journal of Aerosol Science, 18, 131–138, https://doi.org/10.1016/0021-8502(87)90050-4, 1987.

Di Biagio, C., Formenti, P., Balkanski, Y., Caponi, L., Cazaunau, M., Pangui, E., Journet, E., Nowak, S., Andreae, M. O., Kandler, K., Saeed, T., Piketh, S., Seibert, D., Williams, E., and Doussin, J.-F.: Complex refractive indices and single-scattering albedo of global dust aerosols in the shortwave spectrum and relationship to size and iron content, Atmos. Chem. Phys., 19, 15503–15531, https://doi.org/10.5194/acp-19-15503-2019, 2019.

Gonçalves Ageitos, M., Obiso, V., Miller, R. L., Jorba, O., Klose, M., Dawson, M., Balkanski, Y., Perlwitz, J., Basart, S., Di Tomaso, E., Escribano, J., Macchia, F., Montané, G., Mahowald, N. M., Green, R. O., Thompson, D. R., and Pérez García-Pando, C.: Modeling dust mineralogical composition: sensitivity to soil mineralogy atlases and their expected climate impacts, Atmos. Chem. Phys., 23, 8623–8657, https://doi.org/10.5194/acp-23-8623-2023, 2023.

Formenti, P., Caquineau, S., Desboeufs, K., Klaver, A., Chevaillier, S., Journet, E., and Rajot, J. L.: Mapping the physico-chemical properties of mineral dust in western Africa: mineralogical composition, Atmos. Chem. Phys., 14, 10663–10686, https://doi.org/10.5194/acp-14-10663-2014, 2014.

Kandler, K., Schütz, L., Deutscher, C., Ebert, M., Hofmann, H., Jäckel, S., Jaenicke, R., Knippertz, P., Lieke, K., Massling, A., Petzold, A., Schladitz, A., Weinzierl, B., Wiedensohler, A., Zorn, S., and Weinbruch, S.: Size distribution, mass concentration, chemical and mineralogical composition and derived optical parameters of the boundary layer aerosol at Tinfou, Morocco, during SAMUM 2006, Tellus B: Chemical and Physical Meteorology, 61, 32, https://doi.org/10.1111/j.1600-0889.2008.00385.x, 2009.

Lafon, S., Rajot, J.-L., Alfaro, S. C., and Gaudichet, A.: Quantification of iron oxides in desert aerosol, Atmospheric Environment, 38,1211–1218, https://doi.org/10.1016/j.atmosenv.2003.11.006, 2004.

Miffre, A., Cholleton, D., and Rairoux, P.: On the use of light polarization to investigate the size, shape, and refractive index dependence of backscattering Ångström exponents, Optics Letters, 45, 1084, https://doi.org/10.1364/OL.385107, 2020.

Sokolik, I. N. and Toon, O. B.: Incorporation of mineralogical composition into models of the radiative properties of mineral aerosol from UV to IR wavelengths, Journal of Geophysical Research: Atmospheres, 104, 9423–9444, https://doi.org/10.1029/1998JD200048, 1999.

Wagner, R., Ajtai, T., Kandler, K., Lieke, K., Linke, C., Müller, T., Schnaiter, M., and Vragel, M.: Complex refractive indices of Saharan dust samples at visible and near UV wavelengths: a laboratory study, Atmos. Chem. Phys., 12, 2491–2512, https://doi.org/10.5194/acp-12-2491-2012, 2012.

Zhang, Y., Saito, M., Yang, P., Schuster, G., and Trepte, C.: Sensitivities of Spectral Optical Properties of Dust Aerosols to Their Mineralogical and Microphysical Properties, Journal of Geophysical Research: Atmospheres, 129, e2023JD040 181, https://doi.org/10.1029/2023JD040181, 2024

**Replies to the comments from referee #2, Ali Omar**

We would like to start the reply by thanking Ali Omar for his review, and furthermore for the thoughtful, well-structured and comprehensive revision of our manuscript. All the comments and insight are very much appreciated, and have helped to improve the manuscript. We have copied the comments into this document; the comments are in Times New Roman blue font while our answers are in Calibri black font. Line numbers refer to the version of the manuscript with track changes.

**Summary**

This is an excellent paper that will contribute to the reduction of uncertainties in the radiative effects of dust due to insufficient knowledge of dust properties, an in particular the composition of the dust and its effect on optical properties. It highlights the importance of understanding the mineralogical content of dust, particularly the role of iron oxides like hematite, which affect the dust's optical properties. The study uses lidar measurements and atmospheric modeling to explore the relationship between hematite content and dust's optical properties, such as the lidar ratio and Ångström exponent. The findings suggest that while there is a positive correlation between hematite content and certain optical properties, the relationship is complex and influenced by particle size and composition. The study emphasizes the need for further research to better understand these interactions and improve the accuracy of dust's radiative effect estimates in climate models.

**Methodology of Data Selection**

The methodology for selecting data in the dust study is systematic and well-structured, focusing on ensuring that the data is relevant and reliable. Below we acknowledge some strengths and potential areas for improvement:.

**Strengths:**

**Multi-step Approach**: The use of a three-step process (AERONET data, PollyXT measurements, and COSMO-MUSCAT simulations) ensures a comprehensive selection of dust-dominated cases.

**Quality Control**: The focus on data from specific campaigns (JATAC) with rigorous quality control and cross-validation enhances the reliability of the data.

Seasonal Consideration: Selecting data from summer months when Saharan dust transport is most pronounced helps in capturing significant dust events.

**Clear Criteria**: The use of specific criteria (AOT and Ångström exponent) for filtering AERONET data ensures that only relevant dust events are considered.

**Areas for improvement**

**Controlling for variations in intensive properties**

The correlation between lidar parameters and hematite might be influenced by changes in the size distributions of the aerosol layers or the optical thickness of the layers considered for the study. Controlling for these by using layers of comparable optical depths and size distributions will eliminate any influence of the variations in these properties

We appreciate the suggestion since it is important to try to isolate physical parameters that might influence dust intensive optical properties in order to observe the effect that composition has on them. We calculated the AOT per dust layer from lidar particle extinction coefficients at 532nm. Furthermore, we tried clustering dust layers by similar AOT, but we realized that this would not precisely isolate particle size differences within the dust layer, since similar mass loads can arise from different particle size distributions. Nevertheless, we wanted to share the clustering analysis:

[Figure]

Figure 1. Mean values for 20 dust layers separated regarding their AOT values at 532nm: (a-b) dust layers with AOT below 0.2, (c.d) dust layers with AOT above 0.2 and below 0.4. Hematite fraction are shown in the x axis and three intensive optical properties are shown in the y axes per graph. Lidar ratios (a,c: $S_{355}$, b,d:$S_{532}$) are shown in color coded dots. Triangle symbols represent the particle linear depolarization ratio (a,c: $\delta_{355}$, b,d:$\delta_{532}$) and the Ångström exponents are depicted by the 45° tilted "Y" symbols (a,c: ÅE($\beta$)$_{355/532}$, b,d: ÅE($\alpha$)$_{355/532}$). All the standard deviations of the measurements are shown in their error bars. The color coding of the lidar ratio depends on the fine-to-coarse fraction in percentages obtained from the POLIPHON method.

Figure 1 illustrates the separation of dust cases based on their AOT at 532nm, derived from the lidar measurements. Interestingly, the positive correlation with the backscatter-related Ångström exponent is visible for both clusters.

For the size distribution control suggestion, it is important to take into account that the size distribution that can be derived from lidar retrievals would be dependent on a spherical shape assumption, resulting in further uncertainties. We have not done this for the study since we already have size segregation tool via the POLIPHON and the provided fine-to-coarse ratio.

Additionally, we analyzed the simulated particle size distributions (PSDs) for all dust layers. We compared the simulated PSDs to each other, considering the fine-to-coarse groups defined in the manuscript. This comparison revealed significant differences among the simulated PSDs across all groups. However, when we compared the dust layer both mass concentrations with POLIPHON results and the lidar-derived AOTs at 532nm with the simulated AOTs at 550nm, it became clear that the model does not always accurately represent dust layer mass concentrations and mass loading. As a result, some simulated PSDs may also be inaccurate. Given the varying accuracy of the modeling results, we suggest that grouping dust layers based on their simulated PSDs may introduce even greater uncertainty than the size segregating approach proposed in the manuscript.

**Cloud Influence:** While cloud screening is mentioned, the methodology could benefit from a more detailed description of how cloud interference is minimized or accounted for in the data analysis. In particular how does cloud contamination in the data manifest itself in the results.

Thank you for the suggestion, we have added a more detailed explanation of how the manual cloud screening is performed and what could happen if clouds do appear in the retrievals. The modifications can be found in L258-264 in Section 2.3.

**Temporal and Spatial Resolution**: The methodology could discuss the temporal and spatial resolution of the PollyXT and COSMO-MUSCAT data to ensure that the selected cases are representative of broader dust transport patterns.

We have added a more detailed discussion regarding the temporal and resolution differences between the Polly$^{XT}$ and COSMO-MUSCAT in Section 2.1, L130-134. We have further added a sentence on the repercussions of those differences in the results analysis part in Section 3.1.2, L429. Furthermore, we have added a comparison with AOT from AERONET stations across the region reinforcing that the selected cases are representative of the broader dust transport patterns. In regards with the cases being representative of broader dust transport patterns, we have added a sentence while analyzing the results in Section 3.2, L451.

**Model Validation**: While the COSMO-MUSCAT model is used to confirm dust layers, additional validation against independent datasets could strengthen the reliability of the model's outputs.

Thank you, as a result of your suggestion we have added the AERONET AOT at 550nm comparisons that can be found in the Appendix A. The results of this comparison are discussed

in Section 3.2 between L448-459 and the comparison itself can be found in the Appendix A, Figs. A1 and A2.

**Potential Bias:** The focus on specific periods and locations might introduce a bias. Expanding the study to include different times and regions could provide a more comprehensive understanding of dust transport and its properties.

We agree and therefore encourage to broaden this study to have both, more case studies and investigating other dust regions of the world. We now point towards this in the conclusions section in L619-622. Nevertheless, the inclusions of other regions and more case studies are, at the moment, outside of the scope of this particular study.

Overall, the methodology is robust, but incorporating additional validation steps or at least discussing and acknowledging the potential bias and expanding the scope could enhance the study's comprehensiveness and applicability.

Thank you, we have added further discussion on biases and encourage to expanding the scope for subsequent studies.

**Replies to the comments from anonymous referee #3:**

We sincerely thank the referee for their thorough review of our manuscript and for raising important points regarding the need for more physics-based interpretation of our results. We appreciate your insights and the opportunity to clarify and strengthen the physical reasoning underlying our analysis. The line changes mentioned in this document correspond to those in the difference file. We have copied the comments into this document; the referee's comments are in Times New Roman blue font while our answers are in Calibri black font.

I read through the author's response document and the revised manuscript. I recognize that the authors have sufficiently addressed most of the questions/comments raised by the reviewers. However, it appears that the current manuscript does not reflect one comment raised by Reviewer #1, and I think it is the most critical comment for considering the publication of this manuscript.

Throughout the manuscript, there are lack of descriptions on a physics-based interpretation of the results. The authors described the results illustrated in Figs very well, compared the results to previous studies, and showed correlation coefficients to "try to" relate the results with some physical parameters. However, correlation coefficients do not help articulate the causality of phenomena. The causality can be inferred based on a physics-based interpretation, otherwise, the discussion does not go beyond a speculative level.

I also argue the robustness of the analysis method related to Figs. 7-8. Lidar ratio, depolarization ratio, and Angstrom exponents are determined by multiple physical variables, not a single hematite fraction. In particular, because hematite is absorptive (i.e., high imaginary refractive index), the authors should stratify the data with particle sizes. This is because the particle absorptivity is primarily determined by a combination of the imaginary refractive index (i.e., degree of absorption in a particle medium) and particle sizes (i.e., a volume or amount of absorbing medium). In this context, the results shown in Table 3 would be helpful. I suggest the authors provide a physical-basis explanation of the results, in particular, a positive correlation of the lidar ratio at 355 nm relating to the imaginary refractive index of hematite, a negative correlation of the Angstrom exponent (355/532) relating to particle size and spectral characteristics of the imaginary refractive index.

In conclusion, the topic is within the scope of ACP, the manuscript has the potential to be publishable if the authors can address the critical comments raised by Reviewer #1 regarding the lack of physics-based explanations/interpretations. I therefore recommend major revisions to consider this manuscript for publication.

From the previous review, we introduced a more explicit physics-based framework in the *Introduction* to aid in interpreting the results. We appreciate the reviewer's observation that this framework had not yet been adequately integrated into the interpretation of the results. In the revised version of the manuscript, we have now expanded the *Results and their discussion* section to fully integrate the physical basis laid out before.

The physics-based framework in our study centers in the relationship between the imaginary part of the refractive index ($m_i$) and the backscattering coefficient, described by:

$$\beta(\lambda, m) = \int_{D_{max}}^{D_{min}} N(D) \cdot \sigma_\beta(m, D, \lambda) \, dD \quad , \qquad\qquad\qquad (1)$$

where, $\beta$ is the backscattering coefficient, $N(D)$ is the number concentration of particles of diameter $D$, and $\sigma_\beta$ is the backscatter cross-section depending on the complex refractive index ($m = m_R + m_i$), particle size, and wavelength $\lambda$.

Mie theory, which assumes spherical particles, is commonly used to calculate the backscatter cross-section. Although this approach does not capture the full complexity of irregularly shaped mineral dust, it provides a reasonable first-order approximation for assessing how the backscatter coefficient responds to variations in the complex refractive index.

We fully acknowledge that particle non-sphericity plays a significant role in shaping dust optical properties. Previous studies have shown that lidar-derived optical properties modeled using Mie theory can differ substantially from observations (Chipade & Pandya, 2023). To address these limitations, more advance scattering optical models that incorporate non-sphericity have been developed, such as those by Dubovik et al. (2006), Gasteiger et al. (2011), Koepke et al. (2015), and Saito and Yang (2021) and Saito et al. (2021). While these models offer increased physical accuracy, discrepancies between modeled and observed data still persist, and the refinement of dust optical modeling remains an active area of research, including ongoing efforts within our own remote sensing group at TROPOS.

Despite the limitations, our rationale for drawing on Mie-theory-derived trends in this study is grounded in its ability to reproduce key relationships that are consistent across more sophisticated scattering models. This is particularly relevant for intensive optical properties, which are independent of aerosol concentration. For example, Wandinger et al. (2023) showed that the backscatter coefficients dependency on the imaginary part of the refractive index ($m_i$) computed using the spheroidal model of Koepke et al. (2015) closely mirrored those derived from Mie theory. Similarly, Chang et al. (EGUsphere 2024) demonstrated that the negative relationship between $m_i$ and the backscatter coefficient persists regardless of the scattering model employed, whether Mie theory (spheres), the spheroidal model of Dubovik et al. (2006), or the irregular hexahedral model of Saito et al. (2021). To further illustrate this consistency, we include Figure 1 (Fig.5 in Chang et al. (EGUsphere, 2024)) in this answer, which shows that while absolute values differ due to particle shape, the curve of the backscatter coefficient response to increases on $m_i$ remains consistent across models. This convergence supports the robustness of Mie-theory-based trends as a foundation for interpreting the relationships observed in our data.

As part of our physics-based interpretative framework, we refer to Figure 2 in De Leeuw & Lamberts (1987), which presents Mie theory calculations showing that the slope of the $\beta$-to-$m_i$ relationship varies by wavelength, with a steeper decline at 532nm compared to 355nm. This behavior is further explained through size parameter considerations ($\chi = 2\pi r/\lambda$): at 355nm, the larger size parameter results in a more forward-peaked scattering pattern, meaning less energy is initially scattered backward relative to 532nm. Consequently, the baseline backscattering is higher at 532nm. As absorption increases through a higher $m_i$, it dampens internal and surface resonances that enhance backscattering, leading to a more pronounced reduction at 532nm. In contrast, backscatter at 355nm, already lower and less enhanced by resonances is less sensitive to the same increase in absorption. Therefore, under dust particle size regimes and refractive index variations consistent with Di Biagio, et al.

(2019), the steeper decline in backscattering at 532nm reflects its stronger sensitivity to absorption-driven resonance suppression. Although this wavelength-dependent behavior is derived from Mie theory, we consider it robust for interpretation, as the general shape of the backscatter response to increasing $m_i$ is consistent across scattering models. This interpretation is further supported by the behavior of the backscatter-related Ångström exponent shown in Figure 2 in this response (Fig.9 in Chang et al. (EGUsphere, 2024)), which reflects this differential sensitivity of backscattering at the two wavelengths to changes in $m_i$ across scattering models.

Within this framework, we interpret the observed optical variability under the assumption that changes in the modeled hematite fraction are the main driver of variations in the imaginary part of the refractive index. This assumption is supported by the findings of Di Biagio et al. (2019), who demonstrated a positive correlation between hematite content and imaginary refractive index in the UV-Vis spectral range. Their results suggest that even small increases in hematite concentration can significantly influence the optical properties of mineral dust, especially those sensitive to absorption and scattering efficiency.

We use this understanding to interpret the observed relationships between modeled hematite fraction and intensive optical properties. In particular, the backscatter-related Ångström exponent, which depends on the backscatter ratio between 532nm and 355nm reflects the differential sensitivity of each wavelength to increasing $m_i$. As backscatter at 532nm decreases more steeply than at 355nm with increasing $m_i$, the ratio $\frac{\beta_{532}}{\beta_{355}}$ approaches unity. Given the definition of the backscatter-related Ångström exponent:

$$\text{ÅE}(\beta)_{355/532} = -\frac{\ln\left(\frac{\beta_{532}}{\beta_{355}}\right)}{\ln\left(\frac{532}{355}\right)}, \tag{2}$$

this results in an increase $\text{ÅE}(\beta)_{355/532}$ toward zero, as $m_i$ increases for the cases shown in Figure 7 of the manuscript, where all dust plumes analyzed exhibit a negative $\text{ÅE}(\beta)_{355/532}$. The trend in our data is also consistent with Figure 2 in this response (from Chang et al. (EGUsphere)), which shows that the $\text{ÅE}(\beta)_{355/532}$ with increasing $m_i$ behaves in similar manner across the scattering models.

Additionally, we find that the lidar ratios tend to increase with $m_i$. This relationship is largely driven by the decline in the backscatter coefficient, since the extinction coefficient is comparatively less sensitive to changes in $m_i$ (Fig.1). While this relationship is not always apparent across the entire dataset, likely due to this property's sensitivity to particle size distributions, it becomes more apparent in the size-segregated analysis (Figure 8 of the manuscript), where internal particle size variability is reduced and compositional effects are more isolated.

We have updated the manuscript to make these interpretative links more explicit. The added discussion clarifies how variations in $m_i$, linked to hematite content, influence the backscatter coefficient, lidar ratio and Ångström exponent. Furthermore, we have added the slope and intercept of the linear fit in the main results figures in order to facilitate the interpretation of them. The changes can be found in the *Introduction* section between **L60-69** and in the *Results and their discussion* section, particularly sections **3.2.1** and **3.2.2** and in **Figures 7** and **8**.

[Figure]

Figure 1. Dependence of (a) extinction coefficient, (b) backscatter coefficient, and (c) particle depolarization ratio on the imaginary part of the refractive index $m_i$ at 532nm, from Chang et al. (EGUsphere2024). Line styles represent different fixed values of the real part of the refractive index ($m_R$= 1.4, 1.5 and 1.6), as indicated in the legend. Colors denote different scattering models: green for spherical particles using Mie theory, blue for spheroids following Dubovik et al. (2006), and orange for irregular hexahedra according to Saito et al. (2021).

[Figure]

Figure 2. Dependence of the extinction-related Ångström exponent (left) and the backscatter-related Ångström exponent (right) between the 355nm and 532nm wavelengths on the imaginary part of the refractive index $m_i$ from Chang et al. (EGUsphere2024). Line styles represent different fixed values of the imaginary part of the refractive index as indicated in the legend. Colors denote different scattering models as described in Fig.1.

References:

Chang, Y., Hu, Q., Goloub, P., Podvin, T., Veselovskii, I., Ducos, F., Dubois, G., Saito, M., Lopatin, A., Dubovik, O., and Chen, C.: Retrieval of microphysical properties of dust aerosols from extinction, backscattering and depolarization lidar measurements using various particle scattering models, https://doi.org/10.5194/egusphere-2024-2655, 2024.

Chipade, R. A. and Pandya, M. R.: Theoretical derivation of aerosol lidar ratio using Mie theory for CALIOP-CALIPSO and OPAC aerosol models, Atmos. Meas. Tech., 16, 5443–5459, https://doi.org/10.5194/amt-16-5443-2023, 2023.

De Leeuw, G. and Lamberts, C.: Influence of refractive index and particle size interval on mie calculated backscatter and extinction, Journal of Aerosol Science, 18, 131–138, https://doi.org/10.1016/0021-8502(87)90050-4, 1987.

Di Biagio, C., Formenti, P., Balkanski, Y., Caponi, L., Cazaunau, M., Pangui, E., Journet, E., Nowak, S., Andreae, M. O., Kandler, K., Saeed, T., Piketh, S., Seibert, D., Williams, E., and Doussin, J.-F.: Complex refractive indices and single-scattering albedo of global dust aerosols in the shortwave spectrum and relationship to size and iron content, Atmos. Chem. Phys., 19, 15503–15531, https://doi.org/10.5194/acp-19-15503-2019, 2019.

Dubovik, O., Sinyuk, A., Lapyonok, T., Holben, B. N., Mishchenko, M., Yang, P., Eck, T. F., Volten, H., Muñoz, O., Veihelmann, B., Van Der Zande, W. J., Leon, J., Sorokin, M., and Slutsker, I.: Application of spheroid models to

account for aerosol particle nonsphericity in remote sensing of desert dust, Journal of Geophysical Research: Atmospheres, 111, 2005JD006 619, https://doi.org/10.1029/2005JD006619, 2006.

Gasteiger, J., Wiegner, M., Groß, S., Freudenthaler, V., Toledano, C., Tesche, M., and Kandler, K.: Modelling lidar-relevant optical properties of complex mineral dust aerosols, Tellus B, 63, 725–741, https://doi.org/10.1111/j.16000889.2011.00559.x, 2011.

Koepke, P., Gasteiger, J., and Hess, M.: Technical Note: Optical properties of desert aerosol with non-spherical mineral particles: data incorporated to OPAC, Atmos. Chem. Phys., 15, 5947–5956, https://doi.org/10.5194/acp-15-5947-2015, 2015.

Saito, M. and Yang, P.: Advanced Bulk Optical Models Linking the Backscattering and Microphysical Properties of Mineral Dust Aerosol, Geophysical Research Letters, 48, e2021GL095 121, https://doi.org/10.1029/2021GL095121, 2021.

Saito, M., Yang, P., Ding, J., and Liu, X.: A comprehensive database of the optical properties of irregular aerosol particles for radiative transfer simulations, Journal of the Atmospheric Sciences, https://doi.org/10.1175/JAS-D-20-0338.1, 2021.

Wandinger, U., Floutsi, A. A., Baars, H., Haarig, M., Ansmann, A., Hünerbein, A., Docter, N., Donovan, D., Van Zadelhoff, G.-J., Mason, S., and Cole, J.: HETEAC – the Hybrid End-To-End Aerosol Classification model for EarthCARE, Atmospheric Measurement Techniques, 16, 2485–2510, https://doi.org/10.5194/amt-16-2485-2023, 2023.